# Transportability for Bandits
# with Data from Different Environments

**Alexis Bellot,   Alan Malek,   Silvia Chiappa**
Google DeepMind
London, UK
abellot@google.com

## Abstract

A unifying theme in the design of intelligent agents is to efficiently optimize a policy based on what prior knowledge of the problem is available and what actions can be taken to learn more about it. Bandits are a canonical instance of this task that has been intensely studied in the literature. Most methods, however, typically rely solely on an agent's experimentation in a single environment (or multiple closely related environments). In this paper, we relax this assumption and consider the design of bandit algorithms from a combination of batch data and qualitative assumptions about the relatedness across different environments, represented in the form of causal models. In particular, we show that it is possible to exploit invariances across environments, wherever they may occur in the underlying causal model, to consistently improve learning. The resulting bandit algorithm has a sub-linear regret bound with an explicit dependency on a term that captures how informative related environments are for the task at hand; and may have substantially lower regret than experimentation-only bandit instances.

## 1   Introduction

Multi-armed bandits (MABs) constitute one of the most widely used frameworks for modeling decision-making under uncertainty. In this framework, an agent repeatedly takes actions in an environment with the goal of optimizing a desired objective, such as efficiently inferring the action with highest reward or maximizing cumulative rewards in the long run [34]. As in most reinforcement learning problems, there is a substantial amount of exploration involved while the agent learns about reward distributions under different available actions. This process can be costly in many applications; from an ethical perspective, for example, physicians may not risk compromising their patient's health with unknown treatments. It is therefore important to be efficient with experimentation while learning an optimal policy. In the literature, *structured* bandit instances can help navigate the exploration-exploitation trade-off effectively, for example with assumptions on the functional association between action and reward that facilitate estimation such as linear bandits [1, 15, 19] and causal bandits [23, 36, 28, 27, 24, 26, 12, 29, 7].

An alternative approach to alleviate the cost of active experimentation is to consider leveraging prior data or prior experimentation in related environments to inform an agent's decision-making, which leads to the *hybrid learning* paradigm. The expectation (or rather hope) is that informative prior data or prior experimentation can serve to narrow down reward distributions and *warm start* the MAB so as to converge to optimal actions faster and ultimately achieve higher cumulative reward. Current methods can be categorized into multi-task learning [41, 40, 13] and meta-learning [10, 42, 4, 20, 33, 30]. The former aims to solve a prescribed set of related bandit tasks with shared structure, *e.g.* multiple player scenarios with similar reward distributions. The latter considers an arbitrary number of bandit problems whose parameters are sampled $i.i.d.$ from a meta-prior that

37th Conference on Neural Information Processing Systems (NeurIPS 2023).

can be inferred as the agent experiments across the different related tasks[1]. However, both families of methods assume a relatively restricted class of potential changes across environments and rely explicitly on agent experimentation across all environments for learning. If discrepancies across environments are more general, naively leveraging prior data does not necessarily lead to more informative reward distributions or efficiency improvements in a new environment.

As a concrete example, consider a learning scenario in which historical data is available for the design of a clinical trial[2] aiming to determine the optimal level of a hypertension treatment $X$ for Alzheimer's disease $Y$[3]. Alzheimer's aetiology is complex but it is well established that a patient's age $Z$ and blood pressure $W$ contribute to the development of the disease, and so do a number of (typically) unobserved factors, *e.g.*, physical activity levels, socio-economic status, diet patterns, etc. [37] (encoded with a bi-directed dashed edge). Such data can be useful but has to be handled with care, especially if we suspect the clinical trial popula-

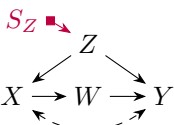

Figure 1: Diagram encoding causal structure and differences across environments.

tion to differ in several aspects from that recorded in historical data. We would expect, for example, age distributions $P(z)$ to differ. Fig. 1 graphically describes this scenario. A naive approach, ignoring the differences across populations, can be sub-optimal. Take an instance where variables $X, Z, W, Y, U \in \{0, 1\}$; their values are decided by functions: $X \leftarrow U, W \leftarrow X, Y \leftarrow Z \oplus (W \cdot U)$; $\oplus$ represents the exclusive-or operator; $U$ is independently distributed in $\{0, 1\}$ but older individuals are historically over-represented $P(Z = 1) = 0.6$ in comparison to the clinical trial population $P(Z = 1) = 0.4$. Historical data suggests that the better policy involves a lower dosage $X = 0$, as $\mathbb{E}[Y \mid do(X = 0)] = 0.6 > 0.5 = \mathbb{E}[Y \mid do(X = 1)]$, which is the opposite of what is optimal for the clinical trial population.

This example shows that differences across environments may be complex, subtle, and non-trivially influence optimal decision-making. Even when one is able to perfectly estimate reward distributions from historical data, the induced policy can still be sub-optimal depending on the location and magnitude of the changes expected across environments. In general, reward distributions will not straightforwardly extrapolate across different environments. In this paper, we attempt to capture this (structural) uncertainty through a causal lens. In the causality literature, this problem appears under the rubric of *transportability theory* [32, 3, 11]; several criteria, algorithms, and estimation methods have been developed for identifying when and how a causal effect can be computed across environments. Our task, in the bandit setting, is to define a learning agent that optimally exploits prior data with knowledge of the potential discrepancies across domains, for *any* given graph. Prior work [43] has considered specific instances, *i.e.* environments defined by the Bow and Instrumental Variables graphs, but a general approach applicable to arbitrary graphs and arbitrary differences across environments is still missing. Our approach is Bayesian, and involves posterior sampling of reward distributions defined by a parameterization informed by the underlying causal structure. Our contribution is to develop a novel bandit algorithm that achieves sub-linear cumulative regret with an explicit dependency on the entropy of an inferred prior, a quantity that implicitly captures the relatedness between environments. The significance of this result is that it guarantees consistent improvements on performance over methods not leveraging prior data. To the best of our knowledge, this is one of the first general attempts to consistently use prior data from related environments in general decision-making scenarios in which causal dependencies can be established.

## 1.1 Preliminaries

We use capital and small letters to denote random variables and their values respectively, *e.g.* $X$ and $x$, and bold capital and small letters to denote sets of variables and their values, *e.g.* $\boldsymbol{X}$ and $\boldsymbol{x}$. The domain of variable $X$ is indicated with $\Omega_X$.

A environment's data generating mechanism is described by a *structural causal model* (SCM) [31, Definition 7.1.1]. A SCM $M$ is a tuple $\langle \boldsymbol{V}, \boldsymbol{U}, \mathcal{F}, P(\boldsymbol{U}) \rangle$, where $\boldsymbol{V}$ is a set of endogenous (observed) variables, $\boldsymbol{U}$ is a set of exogenous latent variables, and $\mathcal{F} = \{f_V\}_{V \in \boldsymbol{V}}$ is a set of functions

---

[1]A more extensive review of related work is given in Appendix A.

[2][17] refer to adaptive clinical trials as the "chief practical motivation [for the design of bandit algorithms]".

[3]For this example, $Y$ a measured biomarker of Alzheimer's disease that acts as a designated reward variable to be maximized.

such that $f_V$ determines values of $V$ taking as argument variables $\boldsymbol{Pa}_V \subseteq \boldsymbol{V}$ and $\boldsymbol{U}_V \subseteq \boldsymbol{U}$, *i.e.* $V \leftarrow f_V(\boldsymbol{Pa}_V, \boldsymbol{U}_V)$. Values of $\boldsymbol{U}$ are drawn from an exogenous distribution $P(\boldsymbol{u})$. We assume the model to be recursive, *i.e.* that there are no cyclic dependencies among the variables, such as to define a distribution $P(\boldsymbol{V})$ over endogenous variables $\boldsymbol{V}$. An intervention or action by an agent on a subset $\boldsymbol{X} \subset \boldsymbol{V}$, denoted by $do(\boldsymbol{x})$, is an operation that fixed values of $\boldsymbol{X}$ to constants $\boldsymbol{x}$, replacing the functions $\{f_X : X \in \boldsymbol{X}\}$ that would normally determine their values. Let $M_{\boldsymbol{x}}$ denote the model induced by action $do(\boldsymbol{x})$. Accordingly, $M_{\boldsymbol{x}}$ induces a corresponding interventional distribution over $\boldsymbol{V}$, denoted $P(\boldsymbol{V}_{\boldsymbol{x}}) := P(\boldsymbol{V} \mid do(\boldsymbol{x}))$. We will consistently use $X, Y \in \boldsymbol{V}$ as designated action and reward variables, respectively.

Causal graphs $\mathcal{G} = (\boldsymbol{V}, \mathcal{E})$ describe the functional associations in an underlying SCM $M$. In particular, we draw a *directed edge* between two variables $V \rightarrow W \in \mathcal{E}$ if $V$ appears as an argument of $f_W$ in $M$, and a *bi-directed dashed edge* between two variables $V \leftarrow\!\text{-}\text{-}\text{-}\text{-}\!\rightarrow W \in \mathcal{E}$ if $\boldsymbol{U}_V \cap \boldsymbol{U}_W \neq \varnothing$, *i.e.* $V$ and $W$ share an unobserved *confounder*. We will use standard family conventions for graphical relationships, *e.g.* parents $pa(\boldsymbol{X})_{\mathcal{G}} := \cup_{X \in \boldsymbol{X}} pa(X)_{\mathcal{G}}$ of a set of nodes $\boldsymbol{X} \subseteq \boldsymbol{V}$ are all nodes in $\mathcal{G}$ with directed edges into elements of $\boldsymbol{X}$. Its capitalized version $Pa$ includes the argument as well, *e.g.* $Pa(\boldsymbol{X})_{\mathcal{G}} := pa(\boldsymbol{X})_{\mathcal{G}} \bigcup \boldsymbol{X}$. We will make use a special clustering of the nodes in $\boldsymbol{V}$ called *c-components* [39]: two nodes are in the same *c*-component $\boldsymbol{C} \subseteq \boldsymbol{V}$ if and only if they are connected by a bi-directed path. *c*-components form a partition over exogenous variables: a *c*-component $\boldsymbol{C} \subseteq \boldsymbol{V}$ is said to cover an exogenous variable $U$ if $U \in \bigcup_{V \in \boldsymbol{C}} \boldsymbol{U}_V$. We denote with $\boldsymbol{C}_U$ the *c*-component covering $U$. As an example, the diagram in Fig. 1 has *c*-components $\{X, Y\}$, $\{W\}$, and $\{Z\}$; and $\boldsymbol{C}_{U_{XY}} = \{X, Y\}$, $\boldsymbol{C}_{U_W} = \{W\}$, and $\boldsymbol{C}_{U_Z} = \{Z\}$. We refer the reader to [31, Chapter 7] for a more detailed review of SCMs.

## 2  Bandits with Transportability

From the agent's perspective, the point of departure with respect to conventional bandit instances is that in addition to the ability to take actions in a deployment environment $\pi^*$, the agent has access to data from one or more related environments $\pi^a, \pi^b, \ldots$, each characterized by SCMs $M^a, M^b, \ldots$. We assume that all environments have the same scope, *i.e.* the same sets $\boldsymbol{V}$ and $\boldsymbol{U}$, but may differ in *any* other aspect. In the transportability literature [3, 32], such structural differences between environments are called domain discrepancies and can be encoded in selection diagrams.

**Definition 1** (Domain Discrepancy). *Let $\pi^a$ and $\pi^b$ be two domains with SCMs $M^a$ and $M^b$. There exists a domain discrepancy between $\pi^a$ and $\pi^b$ if $f_V^a \neq f_V^b$ or $P^a(\boldsymbol{U}_V) \neq P^b(\boldsymbol{U}_V)$ for some $V \in \boldsymbol{V}$.*

**Definition 2** (Selection diagram). *Given domain discrepancy set $\Delta^{a,b} := \{V \in \boldsymbol{V} : f_V^a \neq f_V^b$ or $P^a(\boldsymbol{U}_V) \neq P^b(\boldsymbol{U}_V)\}$ between two domains $\pi^a$ and $\pi^b$ and a causal graph $\mathcal{G}^a = (\boldsymbol{V}, \mathcal{E})$, let $\boldsymbol{S} = \{S_V : V \in \Delta^{a,b}\}$ be called selection nodes. The graph $\mathcal{G}^{a,b} = (\boldsymbol{V} \cup \boldsymbol{S}, \mathcal{E} \cup \{S_V \rightarrow V\}_{S_V \in \mathbf{S}})$ is called selection diagram.*

Selection nodes indicate where structural discrepancies between two environments might take place. The absence of a selection node pointing to a variable represents the assumption that the causal mechanism responsible for assigning values to that variable is identical in both environments. In the clinical trial example, Fig. 1 shows a selection diagram comparing historical and clinical trial environments, denoted $\pi^*, \pi^a$ respectively; the presence of selection node $S_Z$ indicates a potential difference in the assignment of $Z$, *i.e.*, either $f_Z^* \neq f_Z^a$ and / or $P^*(\boldsymbol{u}_W) \neq P^a(\boldsymbol{u}_W)$. On the other hand, the absence of *e.g.* selection node $S_Y$ indicates the assumption $f_Y^* = f_Y^a$ and $P^*(\boldsymbol{u}_Y) = P^a(\boldsymbol{u}_Y)$. With this formalism, the task is to leverage data from related environments in a consistent and efficient manner.

**Definition 3** (Bandits with Transportability). *Let $\pi^*$ denote the deployment environment in which the agent acts. Given samples from $P^a(\boldsymbol{V}), P^b(\boldsymbol{V}), \ldots$ and selection diagrams $\mathcal{G}^{*,a}, \mathcal{G}^{*,b}, \ldots$, in each round $t = 1, \ldots, T$ the agent takes an action $x^{(t)}$ and observes a sample from $P^*(\boldsymbol{V}_{x^{(t)}})$, adjusting its actions to minimize (expected) cumulative regret in $\pi^*$,*

$$\mathbb{E}_{P*} R_T := \sum_{t=1}^{T} \mathbb{E}_{P*} Y_{\tilde{x}} - \mathbb{E}_{P*} Y_{x^{(t)}}, \tag{1}$$

*that compares the optimal intervention $\tilde{x} = \arg\max_{x \in \Omega_X} \mathbb{E}_{P*} Y_x$ with the agent's chosen intervention in each round.*

Quantities such as $\mathbb{E}_{P*}Y_x$ or $P^*(y_x)$ are called *transportability queries* and their estimation with prior data, underlying structural assumptions and their use within active experimentation schemes will be the focus of this paper.

## 2.1 Informative priors for bandits

Before experimentation takes place there is a degree of *unidentifiability* of reward distributions $P^*(y_x)$ depending on causal assumptions and discrepancies between environments. For instance, revisiting the clinical trial example, if age distributions are allowed to vary arbitrarily across environments, values of $P^*(y_x)$ will similarly vary and thus involve a degree of uncertainty[4]. This unidentifiability feature is relevant even without major discrepancies across domains as $P^*(y_x)$ may still be unidentifiable in the presence of unobserved confounders. The Bow graph in Fig. 2 (ignoring the selection node) is a common example.

One may be tempted to conclude that prior data is rarely useful. However, even under multiple discrepancies across environments, causal effects $P^*(y_x)$ are rarely completely unconstrained. In general, causal effects lie in a non-trivial interval $[a, b], 0 < a \leqslant b < 1$. For instance, for the graph $\mathcal{G}^{a,*}$ in Fig. 2 $P^*(y_x)$ can be shown to be contained in $[P^a(x, y), P^a(x, y) + 1 - P^a(x)]$. In particular, with a probabilistic or Bayesian interpretation of unknown quantities in a SCM, that is with an explicit probability measure over SCMs $M \in \mathcal{M}(\mathcal{G}^*)$[5], one

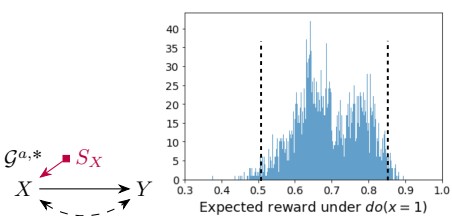

Figure 2: Bow graph and posterior reward density.

can define a *distribution* of reward probabilities $P^*(y_x)$ (or expected rewards) that honestly captures prior uncertainty in their values. For example, given prior samples $\bar{\boldsymbol{v}}^a = \{\boldsymbol{v}^a_{(j)} : j = 1, \ldots, 500\}$ independently drawn from a source distribution $P^a(\boldsymbol{V})$ (and a particular prior model over SCMs, to be discussed in Sec. 2.2), a posterior density over $\mathbb{E}_{P*}Y_x$ can be evaluated and sampled from, see Fig. 2 for an illustration, where theoretical bounds are shown with vertical lines.

From this perspective, the problem of doing inference on reward distributions given prior data and knowledge of structural discrepancies can be formulated as an optimization problem, that is, evaluate,

$$P_{M \sim \mathcal{M}(\mathcal{G}^*)}\left( \mathbb{E}_{P_M}[Y_x] \mid \bar{\boldsymbol{v}} \right), \quad \text{such that} \quad \forall V \notin \Delta^{*,i} : f_V^* = f_V^i, P_{\pi*}(\boldsymbol{u}_V) = P_{\pi^i}(\boldsymbol{u}_V), \quad (2)$$

where $\bar{\boldsymbol{v}} := \{\bar{\boldsymbol{v}}^a, \bar{\boldsymbol{v}}^b, \ldots\}, \bar{\boldsymbol{v}}^i = \{\boldsymbol{v}^i_{(j)} : j = 1, \ldots, n_i\}$ is a set of $n_i$ independent samples from $P^i(\boldsymbol{V})$. In words, the task is to evaluate a distribution over expected rewards under intervention over all deployment domains $\pi^*$ compatible with our knowledge prior to experimenting in $\pi^*$.

It remains a question, however, how to define a model and parameterization of SCMs. Remember that only prior data and selection diagrams are assumed to be available to the researcher; any choices on the distribution of exogenous variables $P(\boldsymbol{U})$ or functional form of deterministic structural assignments $\mathcal{F}$ represent untestable assumptions that are difficult to justify in practice. Going forward we will restrict ourselves to SCMs with *discrete* endogenous variables (while exogenous variables may be arbitrarily defined, *e.g.* continuously-valued with arbitrary probability density functions).

## 2.2 General parameterization of reward distributions

Systems of discrete observables have the distinctiveness of involving a finite number of probabilities of the form $P^*(y_x)$, *i.e.* one for each combination $(x, y)$. Reward distributions $P^*(y_x)$ in any underlying SCM $M$, however complex $P(\boldsymbol{U})$ and $\mathcal{F}$ may be, can logically be equivalently expressed by a corresponding *discrete* SCM $N$ in which $P(\boldsymbol{U})$ is discrete and $\mathcal{F}$ is a discrete mapping between finite spaces [44, 6]. This observation is interesting because it allows us to consistently and uniquely parameterize $P(\boldsymbol{U})$ and $\mathcal{F}$, without untestable choices on their form[6].

---

[4]Formally, we say there exists multiple SCMs $M_1, M_2 \in \mathcal{M}$ with functional dependencies defined by $\mathcal{G}^*$, consistent with prior data and selection diagrams, such that $P_{M_1}(y_x) \neq P_{M_2}(y_x)$.

[5]$\mathcal{M}(\mathcal{G}^*)$ stands for the set of SCMs whose functional dependencies are given by $\mathcal{G}^*$.

[6]A similar reasoning does not apply for continuous endogenous variables that would require continuous exogenous variables and therefore a (untestable) choice of parametric family for all variables.

---

**Algorithm 1** Thompson Sampling with Transportability (tTS)

---

**Input:** Selection diagrams $\{\mathcal{G} * *, a, \mathcal{G}^{*,b}, \dots\}$, prior data $\bar{\boldsymbol{v}} := (\bar{\boldsymbol{v}}^a, \bar{\boldsymbol{v}}^b, \dots)$, decision variable $X$, reward variable $Y$, horizon $T$.
**for** rounds $t = 1, 2, \dots, T$ **do**
  Approximate $P\left(\boldsymbol{\xi}, \boldsymbol{\theta} \mid \bar{\boldsymbol{v}}, \boldsymbol{v}_{x^{(1)}}, \dots, \boldsymbol{v}_{x^{(t-1)}}\right)$
  Sample $\boldsymbol{\xi}^{(t)}, \boldsymbol{\theta}^{(t)} \sim P\left(\boldsymbol{\xi}, \boldsymbol{\theta} \mid \bar{\boldsymbol{v}}, \boldsymbol{v}_{x^{(1)}}, \dots, \boldsymbol{v}_{x^{(t-1)}}\right)$
  $x^{(t)} \leftarrow \arg\max_x \mathbb{E}_{P*}\left[Y_x \mid \boldsymbol{\xi}^{(t)}, \boldsymbol{\theta}^{(t)}\right]$
  Take action $x^{(t)}$ and observe $\boldsymbol{v}_{x^{(t)}}$ in $\pi^*$
**end for**

---

**Corollary 1** (Proposition 2.7. [44]). *For any causal graph $\mathcal{G}$, let $M$ be an arbitrary SCM compatible with $\mathcal{G}$. For any sets $\boldsymbol{Y}, \boldsymbol{X} \subset \boldsymbol{V}$, the interventional distribution $P(\boldsymbol{y_x})$ could be parameterized as*

$$\sum_{\boldsymbol{v} \setminus \{\boldsymbol{x} \cup \boldsymbol{y}\}} \sum_{\substack{u = 1, \dots, d_U, \\ U \in \boldsymbol{U}}} \prod_{V \in \boldsymbol{V}} \mathbb{1}\{\xi_V^{(\boldsymbol{pa}_V, \boldsymbol{u}_V)} = v\} \prod_{U \in \boldsymbol{U}} \theta_u, \tag{3}$$

*where $\theta_u := P(U = u)$ defines exogenous probabilities of discrete variables $U \in \boldsymbol{U}$ with cardinality $d_U = \prod_{V \in \mathbf{C}_U} |\Omega_{\boldsymbol{Pa}(V)}|$; and each $\xi_V^{(\boldsymbol{pa}_V, \boldsymbol{u}_V)}$ is a deterministic mapping between finite domains $\Omega_{\mathbf{Pa}_V} \times \Omega_{\boldsymbol{U}_V} \mapsto \Omega_V$.*

For example, $P^*(y_x)$ in Fig. 1 can be parameterized by

$$\sum_{w, z, u_z, u_w, u_{xy}} \mathbb{1}\{\xi_Y^{(w, z, u_{xy})} = y\} \mathbb{1}\{\xi_W^{(x, u_w)} = w\} \mathbb{1}\{\xi_Z^{(u_z)} = z\} \theta_{u_{xy}} \theta_{u_z} \theta_{u_w}, \tag{4}$$

where, assuming $X, Y, Z, W$ are binary, $\theta_{u_z}$ is a discrete distribution over a finite domain $\{1, 2\}$ since $|\Omega_{U_z}| = |\Omega_{Pa(Z)}| = |\Omega_Z| = 2$, $\theta_{u_w}$ is a distribution over $\{1, \dots, 4\}$ since $|\Omega_{U_W}| = |\Omega_{Pa(W)}| = |\Omega_X| \cdot |\Omega_Z| = 4$, and $\theta_{u_{xy}}$ is distribution over $\{1, \dots, 32\}$ since $|\Omega_{U_{XY}}| = |\Omega_{Pa(X)}| \cdot |\Omega_{Pa(Y)}| = |\Omega_Z| \cdot |\Omega_X| \cdot |\Omega_W| \cdot |\Omega_Z| \cdot |\Omega_Y| = 32$. Corol. 1 guarantees that for any value of $P^*(y_x)$ induced by an arbitrary SCM $M$ there exists a combination of parameters in Eq. (4) that reaches that exact same value. In other words, this parameterization is sufficiently expressive to encode *any* underlying reward distribution.

Parameters that define reward distributions are specific to the deployment environment $\pi^*$. The key observation, however, is that some of them can be inferred with prior data whenever there exists an invariance in $P(\boldsymbol{U})$ or $\mathcal{F}$ across environments as there exists a one-to-one relationship between model parameters and structural features of the underlying SCM. For example, given Fig. 1, the absence of a selection node into $Y$ implies that both functional assignments and exogenous probabilities of $Y$ agree across environments, that is $\xi_Y^* = \xi_Y^a, \theta_{u_Y}^* = \theta_{u_Y}^a$ [7]. Both may thus be approximated with prior data which in turn constraints or informs $P^*(y_x)$ even if other parameters in its expression in Eq. (4) remain unknown. The result is a non-trivial distribution over reward probabilities that may be used to warm-start bandit algorithms, even before any experimentation takes place.

## 2.3 Bandit algorithms

To exploit non-trivial parameter distribution given prior data, a bandit algorithm can be designed to choose actions in proportion to the probability that an intervention leads to highest reward, also known as posterior or Thompson sampling [38, 2]. Specifically, at a particular round $t$ of experimentation in the deployment domain $\pi^*$, *posterior* parameter distributions $P\left(\boldsymbol{\xi}, \boldsymbol{\theta} \mid \bar{\boldsymbol{v}}, \boldsymbol{v}_{x^{(1)}}, \dots, \boldsymbol{v}_{x^{(t-1)}}\right)$ can be evaluated to exactly capture uncertainty given both prior and experimental data up to round $t$. Action $x^{(t)}$ is then chosen according to the one that gives highest reward, *i.e.* $\arg\max_x \mathbb{E}_{P*}\left[Y_x \mid \boldsymbol{\xi}^{(t)}, \boldsymbol{\theta}^{(t)}\right]$, where $(\boldsymbol{\xi}^{(t)}, \boldsymbol{\theta}^{(t)})$ is an independent draw from its posterior distribution. In other words, the agent

---

[7]It will be useful to write $\boldsymbol{\xi} = \{\xi_V^{(\boldsymbol{pa}_V, \boldsymbol{u}_V)} : V \in \boldsymbol{V}, Pa_V \subset \boldsymbol{V}, \boldsymbol{U}_V \subset \boldsymbol{U}\}$ and $\boldsymbol{\theta} = \{\boldsymbol{\theta}_U : U \in \boldsymbol{U}\}$ to group all possible functional assignments and exogenous probability parameters, respectively. We will in general omit environment superscripts on parameters to lighten the notation.

performs natural Bayesian updates based on both the data available in source environments and its own experimentation as interventional samples $\boldsymbol{v}_{x^{(1)}}, \ldots, \boldsymbol{v}_{x^{(t-1)}}$ become available, matching the intuition of most other Thompson sampling bandits in the literature. The full algorithm, called Thompson sampling with Transportability (tTS), is given in Alg. 1.

## 3 Regret guarantees conditional on prior data

We define information-theoretic regret bounds that aim to capture the exploration-exploitation trade-off for Alg. 1 when prior information allows it to infer parts of the environment before experimentation takes place.

Performance in MABs is, to a large extent, intimately related with the agent's uncertainty about which action is optimal, represented by a random variable $\tilde{X} : \Omega_{\boldsymbol{\xi}} \times \Omega_{\boldsymbol{\theta}} \mapsto \Omega_X$ where $\Omega_{\boldsymbol{\xi}} \times \Omega_{\boldsymbol{\theta}}$ defines the space of all models $\mathcal{M}(\mathcal{G}^*)$ consistent with our knowledge of the deployment environment $\pi^*$. For example, $P_{M \sim \mathcal{M}(\mathcal{G}*)}(\tilde{X} = x) = P_{M \sim \mathcal{M}(\mathcal{G}*)}(\mathbb{E}_{P_M}[Y_x] > \mathbb{E}_{P_M}[Y_{x'}], \forall x' \in \Omega_X \backslash \{x\})$ where $P_{M \sim \mathcal{M}(\mathcal{G}*)}$ is a probability mass function defined over $\mathcal{M}(\mathcal{G}^*)$. It is reasonable to assume that one would only choose actions with large regret when it can reduce the uncertainty in $\tilde{X}$ substantially. Following [35, Sec. 5], we define a scalar $\Gamma_t$[8] such that the per-round regret can be bounded by information gain,

$$\mathbb{E}[Y_{\tilde{X}} - Y_{X^{(t)}} \mid \bar{\boldsymbol{V}}_t, \bar{\boldsymbol{v}}] \leqslant \Gamma_t \sqrt{I_{P(\cdot | \bar{\boldsymbol{V}}_t, \bar{\boldsymbol{v}})}\left(\tilde{X}; \boldsymbol{V}_{X^{(t)}}\right)}, \tag{5}$$

where $\bar{\boldsymbol{V}}_t := \{\boldsymbol{V}_{X^{(1)}}, \ldots, \boldsymbol{V}_{X^{(t-1)}}\}$ denotes the agent's history of interactions with $\pi^*$ up to round $t$, and $I_P(X, Y) := \mathcal{D}_{KL}(P(X, Y) \| P(X)P(Y))$ denotes the filtered mutual information defined based on $P$ (where $\mathcal{D}_{KL}$ is the Kullback-Leibler divergence). Expectations, unless otherwise stated, are taken with respect to all random quantities. The following proposition, extended from [35, Prop. 1], shows that the Bayesian regret of an agent acting according to Alg. 1 is sub-linear with a dependency on the entropy of the optimal action $\tilde{X}$.

**Proposition 1.** *Let $R_T$ denote the regret incurred by following Thompson sampling (Alg. 1). For any $T \in \mathbb{N}$ and $\Gamma \geqslant \Gamma_t$, then*

$$\mathbb{E}[R_T \mid \bar{\boldsymbol{v}}] \leqslant \Gamma \sqrt{\mathcal{H}(\tilde{X} \mid \bar{\boldsymbol{v}})T}, \tag{6}$$

*where $\mathcal{H}(\tilde{X} \mid \bar{\boldsymbol{v}})$ is the conditional entropy of $\tilde{X}$ given $\bar{\boldsymbol{v}}$.*

Proofs are given in Appendix C.

This bound is interesting because it cleanly relates the regret with the uncertainty about the optimal action conditioned on prior data. On one extreme, if data from source environments fully characterizes the optimal action, the entropy equals 0, and no further experimentation is required; on the other extreme, if data from source environments have no relationship with the target query, the entropy equals $\log(|\Omega_X|)$, and the bound reverts to conventional worst-case guarantees [35]. The entropy of the optimal action is often not sufficient to capture the information from $\bar{\boldsymbol{v}}$ as $\tilde{X}$ may still have a uniform distribution even though posterior distributions over $(\boldsymbol{\theta}, \boldsymbol{\xi})$ have tightened. In other words, there is additional structure among different reward distributions that is not captured by the entropy of the optimal action. Such a setting can be analyzed with a different assumption on the per round regret that explicitly considers model parameters $(\boldsymbol{\theta}, \boldsymbol{\xi})$ to quantify information gain,

$$\mathbb{E}[Y_{\tilde{X}} - Y_{X^{(t)}} \mid \bar{\boldsymbol{V}}_t, \bar{\boldsymbol{v}}] \leqslant \Gamma_t \sqrt{I_{P(\cdot | \bar{\boldsymbol{V}}_t, \bar{\boldsymbol{v}})}(\boldsymbol{\theta}, \boldsymbol{\xi}; \boldsymbol{V}_{X^{(t)}})} + \epsilon_t. \tag{7}$$

where $\epsilon_t > 0$ is an additional slack term. Accordingly, the following proposition provides an alternative bound using a conditional analogue of [25, Prop. 2].

**Proposition 2.** *Let $R_T$ denote the regret incurred following the policy defined by Alg. 1. For any $T \in \mathbb{N}$, if Eq. (7) holds with $\Gamma \geqslant \Gamma_t$ for all $t$,*

$$\mathbb{E}[R_T \mid \bar{\boldsymbol{v}}] \leqslant \Gamma \sqrt{T I_{P(\cdot | \bar{\boldsymbol{v}})}(\boldsymbol{\theta}, \boldsymbol{\xi}; \boldsymbol{V}_{X^{(1)}}, \ldots, \boldsymbol{V}_{X^{(T)}})} + \sum_{t=1}^{T} \mathbb{E}[\epsilon_t]. \tag{8}$$

---

[8]$\Gamma_t$ is called the information ratio and quantifies the trade-off between incurring low regret and gaining information about the optimal action. $\Gamma_t$ can always be upper-bounded by $|\Omega_X|/2$ [35, Sec. 5]. We provide a derivation of a bound for $\Gamma_t$ for a specific set of environments as an example in Appendix C.3.

This proposition shows that if prior data allows the agent to concentrate $(\boldsymbol{\theta}, \boldsymbol{\xi})$ around some value, additional experimentation does not provide much more information and the regret should be small. In principle, it is possible to get precise per-round regret values by inferring values for $\Gamma_t$ and $\epsilon_t$ through the construction of concentration inequalities for the reward variable, as done by Lu et al. in [25, Lem. 3]. We adapt this result using conditional information-theoretic quantities in the following proposition.

**Proposition 3.** *Fix $\delta > 0$ and choose $\Gamma_t$ such that $\left|Y_x - \mathbb{E}[Y_x \mid \bar{\boldsymbol{V}}_t, \bar{\boldsymbol{v}}]\right| \leqslant \frac{\Gamma_t}{2}\sqrt{I_{P(\cdot|\bar{\boldsymbol{V}}_t, \bar{\boldsymbol{v}})}(\boldsymbol{\theta}, \boldsymbol{\xi}; Y_x)}$ for all $x \in \Omega_X$ simultaneously with probability greater than $1 - \delta$. Then Alg. 1 chooses actions $X^{(t)}$ that satisfy*

$$\mathbb{E}[Y_{\tilde{X}} - Y_{X^{(t)}} \mid \bar{\boldsymbol{V}}_t, \bar{\boldsymbol{v}}] \leqslant \Gamma_t\sqrt{I_{P(\cdot|\bar{\boldsymbol{V}}_t, \bar{\boldsymbol{v}})}(\boldsymbol{\theta}, \boldsymbol{\xi}; \boldsymbol{V}_{X^{(t)}})} + \delta B, \tag{9}$$

*where $B \geqslant 0$ is such that $\sup_{y, y' \in \Omega_Y} y - y' \leqslant B$.*

## 4 Posterior approximations

This section describes a tractable algorithm to evaluate posterior distributions of the form $P(\boldsymbol{\xi}, \boldsymbol{\theta} \mid \bar{\boldsymbol{v}}, \boldsymbol{v}_{x^{(1)}}, \ldots, \boldsymbol{v}_{x^{(t-1)}})$ and its posterior updates when new data is collected. Priors on $\boldsymbol{\xi}, \boldsymbol{\theta}$ may be defined such as to induce tractable conditional distributions that may be used within a Gibbs sampling framework. The Gibbs sampler starts with some initial value for all latent quantities $(\boldsymbol{U}, \boldsymbol{\xi}, \boldsymbol{\theta})$ in our target expected reward

$$\mathbb{E}_{P*}[Y_x] = \sum_{y \in \Omega_Y} y P^*(Y_x = y) = \sum_{\boldsymbol{v}\backslash\{x\}} y \sum_{\substack{u=1,\ldots,d_U \\ U \in \boldsymbol{U}}} \prod_{V \in \boldsymbol{V}\backslash X} \mathbb{1}\{\xi_V^{(\boldsymbol{pa}_V, \boldsymbol{u}_V)} = v\} \prod_{U \in \boldsymbol{U}} \theta_u, \tag{10}$$

and samples each one iteratively using their conditional distributions, each parameter conditioned on the current values of the remaining terms in the parameter vector and the available data [16]. As mentioned, what data point carries information about which parameters depends on the structural differences between environments.

**Prior.** For every $V \in \boldsymbol{V}, \forall \boldsymbol{pa}_V, \boldsymbol{u}_V$, the functional assignment parameters $\xi_V^{(\boldsymbol{pa}_V, \boldsymbol{u}_V)}$ are drawn uniformly in the discrete domain $\Omega_V$. For every $U \in \boldsymbol{U}$, exogenous probabilities $\boldsymbol{\theta}_U$ with dimension $d_U = \prod_{V \in \mathbf{C}_U} |\Omega_{Pa(V)}|$ are drawn from a prior Dirichlet distribution (here chosen for conjugacy with the categorical distribution of $\boldsymbol{U}$), $\boldsymbol{\theta}_U = (\theta_1, \ldots, \theta_{d_U}) \sim Dir(\alpha_1, \ldots, \alpha_{d_U})$, with hyperparameters $\alpha_1, \ldots, \alpha_{d_U}$.

**Posterior.** In a particular round $t$, the Gibbs sampler iterates over the following steps.

1. *Sample $\boldsymbol{u}$.* We start by sampling a corresponding exogenous latent variable for each observed sample $\boldsymbol{v}^{(n)} \in (\bar{\boldsymbol{v}}, \boldsymbol{v}_{x^{(1)}}, \ldots, \boldsymbol{v}_{x^{(t-1)}}), n = 1, \ldots, t-1+\sum_i n_i$. Let $(\bar{\boldsymbol{u}}, \boldsymbol{u}_{x^{(1)}}, \ldots, \boldsymbol{u}_{x^{(t-1)}})$ denote the corresponding set of samples of $\boldsymbol{U}$. Exogenous variables $\boldsymbol{U}^{(n)}$ are mutually independent given $\boldsymbol{V}^{(n)}, \boldsymbol{\xi}, \boldsymbol{\theta}$ and thus we can sample each separately using the conditional

$$P(\boldsymbol{u}^{(n)} \mid \boldsymbol{v}^{(n)}, \boldsymbol{\xi}, \boldsymbol{\theta}) \propto P(\boldsymbol{u}^{(n)}, \boldsymbol{v}^{(n)} \mid \boldsymbol{\xi}, \boldsymbol{\theta}) = \prod_{V \in \boldsymbol{V}} \mathbb{1}\{\xi_V^{(\boldsymbol{pa}_V^{(n)}, \boldsymbol{u}_V^{(n)})} = v^{(n)}\} \prod_{U \in \boldsymbol{U}} \theta_u. \tag{11}$$

2. *Sample $\boldsymbol{\xi}$.* Similarly, for fixed $\boldsymbol{pa}_V, \boldsymbol{u}_V$, parameters $\xi_V^{(\boldsymbol{pa}_V, \boldsymbol{u}_V)}$ are mutually independent given $\bar{\boldsymbol{v}}, \boldsymbol{v}_{x^{(1)}}, \ldots, \boldsymbol{v}_{x^{(t-1)}}, \bar{\boldsymbol{u}}, \boldsymbol{u}_{x^{(1)}}, \ldots, \boldsymbol{u}_{x^{(t-1)}}, \boldsymbol{\theta}$. As mentioned, each parameter is updated with the subset of $(\bar{\boldsymbol{v}}, \boldsymbol{v}_{x^{(1)}}, \ldots, \boldsymbol{v}_{x^{(t-1)}}, \bar{\boldsymbol{u}}, \boldsymbol{u}_{x^{(1)}}, \ldots, \boldsymbol{u}_{x^{(t-1)}})$ associated with environments in which the functional assignment of $V$ is invariant across source and deployment environments. As they represent a mapping between variables, its conditional distribution is given by $P(\xi_V^{(\boldsymbol{pa}_V, \boldsymbol{u}_V)} = v \mid \bar{\boldsymbol{v}}, \boldsymbol{v}_{x^{(1)}}, \ldots, \boldsymbol{v}_{x^{(t-1)}}, \bar{\boldsymbol{u}}, \boldsymbol{u}_{x^{(1)}}, \ldots, \boldsymbol{u}_{x^{(t-1)}}) = 1$ if there exists a (relevant) sample $(v^{(n)}, \boldsymbol{pa}_V^{(n)}, \boldsymbol{u}_V^{(n)})$ that fixes the mapping $\boldsymbol{pa}_V^{(n)}, \boldsymbol{u}_V^{(n)} \mapsto v^{(n)}$. Otherwise, $P(\xi_V^{(\boldsymbol{pa}_V, \boldsymbol{u}_V)} = v \mid \bar{\boldsymbol{v}}, \boldsymbol{v}_{x^{(1)}}, \ldots, \boldsymbol{v}_{x^{(t-1)}}, \bar{\boldsymbol{u}}, \boldsymbol{u}_{x^{(1)}}, \ldots, \boldsymbol{u}_{x^{(t-1)}})$ is sampled uniformly from a discrete distribution over $\Omega_V$.

3. *Sample $\boldsymbol{\theta}$.* Fix $U \in \boldsymbol{U}$. By conjugacy of Dirichlet distributions with the categorical distribution, its conditional distribution given all other quantities is given by a Dirichlet distribution $\boldsymbol{\theta}_U \mid$

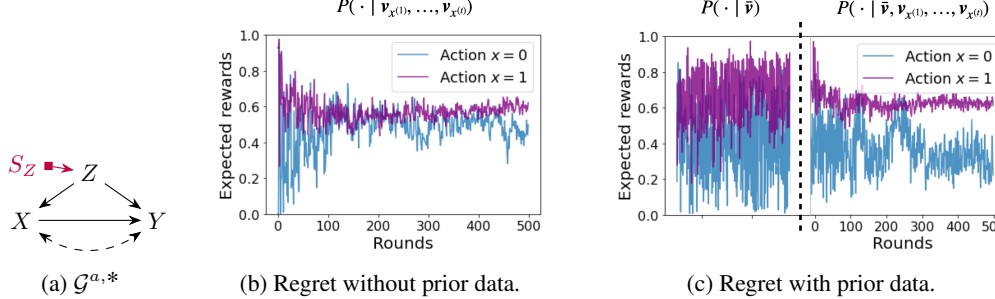

(a) $\mathcal{G}^{a,*}$      (b) Regret without prior data.      (c) Regret with prior data.

Figure 3: Performance figures related to Experiment 1.

$\bar{\boldsymbol{v}}, \boldsymbol{v}_{x^{(1)}}, \ldots, \boldsymbol{v}_{x^{(t-1)}}, \bar{\boldsymbol{u}}, \boldsymbol{u}_{x^{(1)}}, \ldots, \boldsymbol{u}_{x^{(t-1)}} \sim Dir\left(\beta_1, \ldots, \beta_{d_U}\right)$ where $\beta_j := \alpha_j + \sum_n \mathbb{1}\{u^{(n)} = u_j\}$, and, similarly, $n$ iterates over the samples $(\bar{\boldsymbol{u}}, \boldsymbol{u}_{x^{(1)}}, \ldots, \boldsymbol{u}_{x^{(t-1)}})$ associated with the subset of environments in which exogenous probabilities match the deployment environment.

This procedure eventually forms a Markov chain with the invariant distribution $P(\boldsymbol{u}, \boldsymbol{\xi}, \boldsymbol{\theta} \mid \bar{\boldsymbol{v}}, \bar{\boldsymbol{v}}_{x^{(1)}}, \ldots, \bar{\boldsymbol{v}}_{x^{(t-1)}})$. We plug in one of these samples $\boldsymbol{\xi}, \boldsymbol{\theta} \sim P(\boldsymbol{\xi}, \boldsymbol{\theta} \mid \bar{\boldsymbol{v}}, \bar{\boldsymbol{v}}_{x^{(1)}}, \ldots, \bar{\boldsymbol{v}}_{x^{(t-1)}})$ into Eq. (10) for different $x$ to choose the next action $x^{(t)}$. This sample initializes the chain for $P(\boldsymbol{u}, \boldsymbol{\xi}, \boldsymbol{\theta} \mid \bar{\boldsymbol{v}}, \bar{\boldsymbol{v}}_{x^{(1)}}, \ldots, \bar{\boldsymbol{v}}_{x^{(t)}})$ in round $t$ of Alg. 1.

So far, we have described algorithms, approximations, and regret guarantees that pre-suppose the correct specification of causal and selection diagrams across multiple domains. Some degree of misspecification, however, can be tolerated without voiding guarantees on performance improvements. We discuss more details in Appendix B.1.

## 5 Experiments

We evaluate the proposed approach on several synthetic scenarios inspired by the literature on clinical trials and advertising. We compare Thompson sampling with additional data sources (tTS, Alg. 1) with Thompson sampling with uninformative priors (TS) [38], a KL-UCB [9] algorithm with uninformative priors (UCB), and as a baseline also include the algorithm that chooses actions uniformly at random (Uniform)[9]. For all algorithms, we measure their regrets $R_T$, averaged over 10 repetitions. Details on all data generating mechanisms and a discussion on mis-specification and limitations of the proposed approach can be found in Appendix D and Appendix B, respectively.

**Experiment 1.** We start by evaluating the usefulness of prior data by comparing learned distributions of expected reward with and without access to prior data. We consider a bandit problem with action, reward and contextual variables $X, Y, Z \in \{0, 1\}$, respectively, characterized by Fig. 3a in which 1000 prior data samples are given from an environment $\pi^a$ that differs in the causal assignment of $Z$ in comparison with the deployment environment $\pi^*$. Specifically, with this model,

$$P^*(y_x) = \sum_{z, u_z, u_{xy}} \mathbb{1}\{\xi_Y^{(x,z,u_{xy})} = y\} \mathbb{1}\{\xi_Z^{(u_z)} = z\} \theta_{u_z} \theta_{u_{xy}}, \tag{12}$$

where $(\xi_Y, \theta_{u_{xy}})$ are invariant across environments while $(\xi_Z, \theta_{u_z})$ are specific to the deployment environment. We start by considering Fig. 3b that gives samples from $\mathbb{E}_{P*}[Y_x] \mid \boldsymbol{v}_{x^{(1)}}, \ldots, \boldsymbol{v}_{x^{(t)}}$ as a function of experimentation rounds $t$, that is without making use of prior data $\bar{\boldsymbol{v}}$. Distributions of expected rewards under action $x = 0$ and $x = 1$ overlap substantially until round $t = 300$ at which point $x = 1$ is inferred to lead to higher expected rewards. Fig. 3c gives a similar plot with the exception that the left part of the plot gives prior samples $\mathbb{E}_{P*}[Y_x] \mid \bar{\boldsymbol{v}}$ illustrating the shape of the expected reward distribution learned from prior data only. In particular, we observe $\mathbb{E}_{P*}[Y_{x=1}] \mid \bar{\boldsymbol{v}}$ concentrated in the interval $[0.3, 0.9]$ and $\mathbb{E}_{P*}[Y_{x=0}] \mid \bar{\boldsymbol{v}}$ concentrated in the interval $[0.1, 0.8]$. The

---

[9]In contrast to related work, causal bandit algorithms are designed for *single* environments with *multiple* intervention targets, meta-learning bandit algorithms require *experimentation* in multiple related environments, while in the setting considered in this paper, data from source environments are given as a batch. Both sets of methods thus focus on a different class of problems. See Appendix A for more details.

agent starts experimentation at round $t = 0$ with this prior and from then onward expected reward samples are drawn from $\mathbb{E}_{P*}[Y_x] \mid \bar{\boldsymbol{v}}, \boldsymbol{v}_{x^{(1)}}, \ldots, \boldsymbol{v}_{x^{(t)}}$ as a function of experimentation round $t$. The bandit algorithm with prior data is remarkably more efficient, being able to determine $x = 1$ as the superior action after only 80 rounds of experimentation. Overall, prior data leads the bandit algorithm to pull the optimal arm in $99\%$ of time versus $93\%$ of the time without prior data.

**Experiment 2.** We revisit our introductory example to quantify the benefit of leveraging historical patient data from various hospitals. In this example, the objective is to infer the optimal level of hypertension medication $X$. We are given a choice among 5 different levels, *i.e.* $|\Omega_X| = 5$, and wish to increase the probability of the presence of a beneficial biomarker $Y$, *i.e.* $|\Omega_Y| = 2$, in the clinical trial population $\pi^*$ given that we have prior observational data in a different hospital $\pi^a$. The selection diagram describing this causal protocol is given in Fig. 1. Regret comparisons for all algorithms are given in Fig. 4 (LHS). We observe a significant gain in performance by tTS that chooses the optimal intervention in $67\%$ of the rounds in contrast with $35\%$ of the rounds for TS, and $28\%$ of the rounds for UCB (and $17\%$ for an algorithm choosing interventions at random).

We use this example also to illustrate empirically the dependence between regret and prior entropy shown in Prop. 1 (RHS). For this, we consider different prior beta distributions for $P^*(W = 1 \mid x)$, specifically with increasing standard deviations around the true value of $P^*(W = 1 \mid x)$ for each $x \in \Omega_X$. A larger standard deviations implies a less informative prior and higher entropy of the random variable $\tilde{X}$ that denotes the optimal action. The

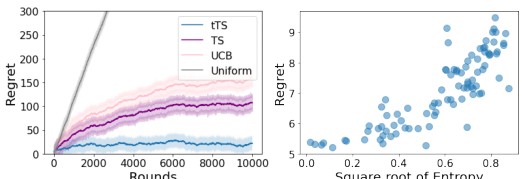

Figure 4: Performance figures related to Experiment 2.

entropy of $\tilde{X}$ takes values in the interval $[0.2, 1.4]$ for assumed Beta priors for $P^*(W = 1 \mid x)$ with standard deviations in the interval $[0.001, 0.1]$. Fig. 4 (RHS) demonstrates empirically the influence of the entropy of $\tilde{X}$ on the expected cumulative regret given in Prop. 1. In particular, narrower, more informative priors lead to better regret.

**Experiment 3.** This example considers an advertiser seeking to optimize which ads to show visitors on a particular website. For each visitor, we choose one out of a collection of 6 ads $X$, $|\Omega_X| = 6$, some of which will be more engaging than others, to ultimately optimize whether a user clicks $Y \in \{0, 1\}$. Each ad has some theoretical but unknown click-through-rate $P^*(y_x)$.

In this example, we assume access to 500 data points from an ad recommendation system used on a different website, *i.e.* a different environment $\pi^a$. There, the effect of an add $X$ on the number of clicks $Y$ is confounded by the user's age $A$, (here categorized into old and young such that $\Omega_A = \{0, 1\}$ and $|\Omega_A| = 2$) and the user's product preferences level $W$, which interacts with current ad-recommendation system through a user's

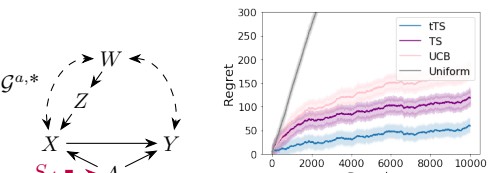

Figure 5: Performance figures related to Experiment 3.

browsing history $Z$ and location (not observed and therefore represented with a bi-directed edge between $W$ and $X$). Moreover, in this example, the relationship between $W$ and $Y$ is itself confounded by unobserved factors. The population visiting the website of interest $\pi^*$, where the MAB will be deployed, is known to agree on all causal components with $\pi^a$ except on the distribution of age $A$. This causal protocol as well as regret comparisons for this example are shown in Fig. 5. We observe noticeable improvements in regret with prior data and knowledge of structural differences as tTS substantially improves over algorithms agnostic of prior data.

# 6 Conclusions

This paper investigated the problem of improving the efficiency of multi-armed bandits using batch data from related environments. As source environments might differ, some knowledge of structure and (potential) discrepancies are necessary to extrapolate consistently. This paper demonstrated that knowledge of selection diagrams that encode causal influence as well as potential discrepancies

across source and target environments, without, however, an explicit specification of functional form and distributions, is sufficient to consistently define an informative prior over reward distributions using data from arbitrary environments. The resulting algorithm guarantees improvements in regret in comparison to algorithms agnostic of prior data. To our knowledge, this serves as one of the first principled approaches to consistently leverage prior data in the context of bandits and we hope it can pave the way for developing more general transfer learning methods in reinforcement learning.

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

# Appendix for "Transportability for Bandits with Data from Different Environments"

This Appendix includes

- Related work in Appendix A.
- Discussion on limitations and analysis of misspecification in Appendix B.
- Proofs in Appendix C.
- Details on the data generating mechanisms Appendix D.
- Details on the Gibbs sampler in Appendix E.

## A    Related work

In the bandit literature, problems of transfer learning in which existing experience and knowledge is used to improve the performance of an agent appear under the rubric of multi-task learning and meta-learning. Typically, in the context of multi-task learning the agent aims to solve a prescribed set of related bandit tasks with shared structure. For example, [41] consider the setting in which multiple players interact in an environment with the property that each player has slightly different associated reward distributions. [40] extend this approach to the Thompson sampling algorithm with an assumption that reward distribution between players are close but not identical. Similar ideas can be found in the contextual case in which, *e.g.* [13] propose to leverage similarities in contexts for different arms and improve prediction of reward distributions from contexts. In the context of meta-learning, the agent is designed to work well on an arbitrary number of tasks from a common environment (*i.e.* sampled from a prescribed distribution), relying on already completed tasks from the same environment. For example, [4, 20, 33] assume a hierarchical bandit structure in which parameters governing multiple bandit instances are sampled $i.i.d.$ from a meta-prior. The authors demonstrate that updates on the meta-prior from one instance can benefit other instances and that regret bounds over a sequence of instances can be established. [30] adopt a similar setting to infer a Lipschitz continuity constant which can be used to derive scale free regret bounds. Others, *e.g.* [10, 42], have assumed that structural parameters of each instance can be decomposed into a shared component and an instance-specific component, and learned with high-dimensional regularization schemes.

Existing methods adopt a relatively wide range of assumptions on the "relatedness" of different bandit instances or players that include, as surveyed above, the existence of a meta-distribution across bandit instances, assumptions on the pairwise similarities in the reward distributions of different players, assumptions on the decomposition of bandit parameters, and assumptions on the distribution of contextual variables and their association with rewards. The environment and causal structure typically remain invariant across bandit instances, and only small and specific set of changes in distribution are allowed across bandit instances.

The proposed approach aims to tackle a more general setting in which we allow for arbitrary changes in the causal mechanisms underlying each environment as long as their location can be established and encoded in selection diagrams, *i.e.* graphs that describe the difference in structure across domains without constraining their form. In the causality literature, these analyses fall under the transportability umbrella. Several authors have demonstrated the power of this approach to identify causal effects, predict counterfactual distributions and, more generally, make inference across different environments [3, 11, 32, 5]. In the bandit literature, there is also an extensive literature on various ways of exploiting knowledge of a causal graph for a single environment, known as causal bandits [23, 36, 28, 27, 24, 26, 12, 29, 7, 18]. Within this literature the causal graph specifies a dependency structure between different actions that allows an agent to identify redundant actions and ultimately more efficient exploration, improving regret bounds by a multiplicative factor. Practical examples of the use of causal graphs (within decision-making problems) are prevalent in the contexts of clinical trials, healthcare, and advertising. We note, in particular, the examples in Figs. 2, 3, 4, in [22] that encode the design of case-cohort studies and clinical trials used in the MORGAM study [14] using causal graphs for the estimation of causal effects. Separately, [21] consider the

International Stroke Trial and the known causal associations between relevant features for policy optimization. In the context of advertising, [8] describes the use of causal methods with several detailed examples. Specifically, Figs. 3, 4, and 6 in [8] show examples of causal graphs that may be defined for particular computational advertising applications. See also [5] for a bandit algorithm leveraging the computational advertising graphs in [8], and[36, Sec. 5.2] for similar causal treatments also in the context of advertising. So far, however, all bandit algorithms exploiting causal knowledge have been studied in the "single environment" setting without access to prior data. Variations among existing proposals are due to the class of graphs or the extent of knowledge of the graph, *e.g.* with or without knowledge of the full causal graph, with or without unobserved confounding, etc. One may interpret the proposed approach as extending the causal bandit formalism to problems in which data from multiple different environments is available prior to an agent experimenting in a target environment, all of which are, as in the causal bandit literature, described by underlying causal models.

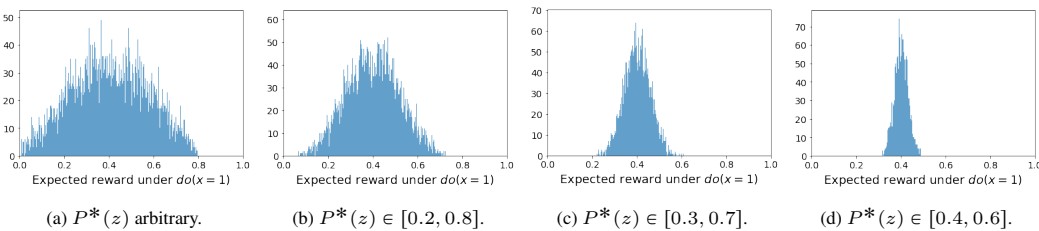

(a) $P^*(z)$ arbitrary.       (b) $P^*(z) \in [0.2, 0.8]$.      (c) $P^*(z) \in [0.3, 0.7]$.      (d) $P^*(z) \in [0.4, 0.6]$.

Figure 6: Prior expected reward distributions under misspecification of $Z$.

## B  Limitations and further analysis

Knowledge of structure may detract from the appeal of active experimentation and might limit the extent to which the proposed approach can be applied. Although in practice several techniques can be used to alleviate this problem such as causal discovery or considering multiple causal graphs as potential causal explanations for the phenomenon of interest, significant domain knowledge will in general be necessary for consistently using prior data. This observation holds for causal bandits more generally.

### B.1  Misspecification

Some degree of misspecification, however, can be handled without voiding guarantees on performance improvements, especially when the assumed causal graphs and selection diagrams form *super-models* for the underlying phenomenon of interest.

Starting with causal graphs: a causal graph $\tilde{\mathcal{G}}^{a,b}$ defined as a graph $(\mathbf{V}, \tilde{\mathcal{E}})$ is said to be a super-model of a causal graph $\mathcal{G} = (\mathbf{V}, \mathcal{E})$ $\mathcal{E} \subseteq \tilde{\mathcal{E}}$. For any causal graph $\mathcal{G}$ it holds that adding an edge, creating a new graph $\tilde{\mathcal{G}}$, creates a larger set of SCMs $\mathcal{M}(\tilde{\mathcal{G}})$ that subsumes $\mathcal{M}(\mathcal{G})$, *i.e.* $\mathcal{M}(\tilde{\mathcal{G}}) \subset \mathcal{M}(\tilde{\mathcal{G}})$, and as a consequence any SCM with non-zero mass under $P_{\mathcal{M}(\mathcal{G})}$ also has non-zero mass under $P_{\mathcal{M}(\tilde{\mathcal{G}})}$. This implies that prior distributions $P_{M \sim \mathcal{M}(\tilde{\mathcal{G}})}(\,\mathbb{E}_{P_M}[Y_x] \mid \bar{\boldsymbol{v}}\,)$ remain valid whenever the causal graph $\tilde{\mathcal{G}}$ is a super-model of the true underlying graph $\mathcal{G}$ as the space of models defined by $\mathcal{G}$ is included in the space of models defined $\tilde{\mathcal{G}}$. For example, consider a scenario in which source and target domains are equal with an underlying causal graph given by $\mathcal{G} := \{X \to Y\}$. Prior to experimentation with knowledge of the graph and source data distribution $P(x, y)$, reward probabilities under agent intervention can be uniquely computed, *i.e.* $P(y \mid do(x)) = P(y \mid x)$. However, if we are unsure about the presence of an unobserved confounder between $X$ and $Y$, we may instead consider a super-model $\mathcal{G} := \{X \to Y, X \leftarrow\!\!\dashrightarrow Y\}$ under which reward probabilities under agent intervention can be determined to be distributed in the interval $[P(x, y), P(x, y) + 1 - P(x)]$, that, in particular, includes the true value $P(y \mid do(x)) = P(y \mid x)$. Inference based on $\tilde{\mathcal{G}}$ instead of $\mathcal{G}$ leads to less informative reward probabilities but remains correct and informative with respect to methods not leveraging prior data as $0 \leqslant P(x, y) \leqslant P(y \mid do(x)) \leqslant P(x, y) + 1 - P(x) \leqslant 1$.

A similar reasoning can be applied to selection diagrams and selection nodes, as the latter only indicates a *potential* discrepancy across domains. A selection diagram $\tilde{\mathcal{G}}^{a,b}$ defined as a graph $(\mathbf{V} \cup \tilde{\mathbf{S}}, \tilde{\mathcal{E}})$ is said to be a super-model of a selection diagram $\mathcal{G}^{a,b} = (\mathbf{V} \cup \mathbf{S}, \mathcal{E})$ if $\mathbf{S} \subseteq \tilde{\mathbf{S}}$ and $\mathcal{E} \subseteq \tilde{\mathcal{E}}$. In words, super-models can have more edges or selection nodes. If a researcher is unsure of whether a causal mechanism differs or not, they may still conduct correct

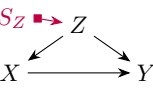

Figure 7: $\mathcal{G}^{a,*}$

inference by being conservative and assuming that a discrepancy exists. Under these types of misspecification, where assumed selection diagrams represent super-models of the underlying data generating process all claims and propositions hold and we can guarantee to never under-perform methods that do not consider prior data or their structure. For example, consider a system with binary variables $X, Y, Z \in \{0, 1\}$ whose causal association and differences across domains are given by Fig. 7. In reality, target and source environments coincide and, with access to batch data from the source environment $\pi^a$, one could uniquely identify expected rewards by computing $\mathbb{E}_{P^*} Y_x = \sum_y y \sum_z P^a(y \mid x, z) P^a(z)$. However, we suspect some degree of discrepancy in the

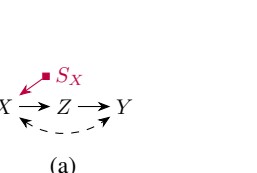
(a)

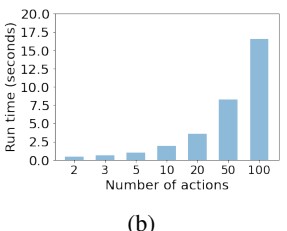
(b)

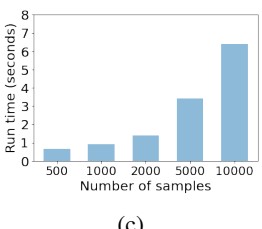
(c)

Figure 9: Run time experiments.

causal assignment of $Z$ across environments, and will consider inference of expected rewards under various degrees of discrepancy, *e.g.* one may explicitly assume that either $P*(z)$ is arbitrary or for example assume $P*(z) \in [a, b]$. We illustrate various such scenarios in Fig. 6. Each panel, from right to left increases allows for a greater level of uncertainty that results in a correspondingly wider distribution of expected reward prior to experimentation. The corresponding regret of bandit algorithms that incorporate these bounds on $Z$ in $P*$ to warm start the distribution of expected regret prior to experimentation are given in Fig. 8. We observe that the wider the prior the higher the corresponding regret, which illustrates how degrees of prior knowledge influence regret.

Using the same reasoning, it holds that if instead the researcher misses edges or incorrectly assigns priors on the causal mechanisms to be expected in the target domain, prior inference will be incorrect in general. For example, if distributions of expected reward in Fig. 6 do not cover the true expected reward, then a bandit algorithm will need to correct for such a biased prior and we can expect cumulative rewards to be lower than those of an algorithm that ignores prior data and structure. The extent to which the corresponding regret of a bandit algorithm incorrectly making assumptions on structure and discrepancies across environments will depend on the underlying causal graph and functional associations between variables.

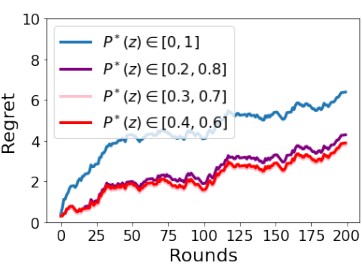

Figure 8: Regret under misspecification.

Most of the discussion in the main body of this paper has assumed knowledge of a discrepancy between environments without considering its magnitude, although the previous example shows that knowledge of bounds or distributions of some of the variables that differ in the target domain can be incorporated. It is worth noting that other kinds of domain knowledge may be available, such as in the functional form of an association in the target domain, specific magnitudes of differences expressed within such a functional form, etc. It would be an interesting task to consider how to use that information optimally.

### B.2    Run time evaluations

Reward distributions $P*(y_x)$ induced by any underlying SCM can be expressed by a corresponding discrete SCM in which $P(\boldsymbol{U})$ is discrete and $\mathcal{F}$ is a discrete mapping between finite spaces. The complexity of the corresponding parameterization depends on the underlying selection diagram and dimensionality of observed variables.

In particular, in a given iteration of the Gibbs sampler, posterior updates are done for each parameter separately so that computational time is proportional to the parameter count, approximately, which in turn is determined by the cardinality of variables as well as the structure of the graph. For a fixed graph, assuming that each update requires a small constant amount of time to compute, we could therefore establish analytically how computational time scales with the cardinality of variables. For arbitrary graphs, an analytical result is in general more involved as the parameter count increases differently depending on the local structure of each variable. As an example for illustration, consider the graph in Fig. 9a where $X$ is an action variable, $Z$ is a contextual variable, and $Y$ is a reward variable. Following the parameterization in Corol. 1, the cardinality of parameters is defined as follows: $|\boldsymbol{\theta}_U| = |\Omega_X| \cdot |\Omega_Z| \cdot |\Omega_Y|, |\boldsymbol{\xi}_X| = |\Omega_X| \cdot |\Omega_Z| \cdot |\Omega_Y|, |\boldsymbol{\xi}_Z| = |\Omega_X|, |\boldsymbol{\xi}_Y| = |\Omega_X| \cdot |\Omega_Z| \cdot |\Omega_Y| \cdot |\Omega_Z|.$

We would expect run time to increase linearly with the cardinality of variables $X, Y$ and to increase "slower than quadratically" with the cardinality of $Z$. Fig. 9b gives run time evaluations of the proposed bandit algorithm using 1000 rounds of experimentation as a function of the number of actions. Fig. 9c evaluates the influence of the size of prior data on run time by plotting the time required for drawing 1000 samples from the prior $P(\cdot \mid \bar{v})$ with increasing prior data size (in addition to 1000 samples Gibbs samples that are discarded as burn-in). For both Figs. 9b and 9c we observe run time to scale approximately linearly with the cardinality of the action space and sample size.

# C  Theoretical results

## C.1  Preliminaries

**Counterfactuals and optimal actions**  We have defined in Sec. 1.1 the fact that intervening in an SCM results in a new model that differs in the causal mechanisms subject to interventions. In this section, we extend this notion to consider models derived from interventions that replace a causal mechanism with another function, not necessarily a constant. This gives consistent definitions for counterfactual random variables of the form $Y_X$ that appear in Sec. 3.

**Definition 4** (Intervened model). *Let $\mathcal{M}$ be an SCM, $\hat{U} \subseteq U$, $X \in V$, and $\hat{X} : \hat{U} \to \Omega_X$ a function. Then, $\mathcal{M}_{\hat{X}}$, called the intervened model of $\mathcal{M}$ subject to $\hat{X}$, is identical to $\mathcal{M}$, except that the function $f_X$ is replaced with the function $\hat{X}$.*

Uncertainty in $\mathcal{M}_{\hat{X}}$ is similarly encoded by the distribution $P(U)$ which when averaged over leads to random variables $Y_{\hat{X}}$ for $Y \in V$ that describes the random variable $Y$ under a model $\mathcal{M}_{\hat{X}}$. This definition is tightly related to that of potential outcome as define in Sec. 1.1, but the former explicitly allows for interventions that do not necessarily fix the variable $X$ to a constant value. Probabilities in this model can be computed using the chain rule

$$P(Y_X = y) = \int_{\Omega_X} P(Y_x = y \mid X = x)P(X = x)dx. \tag{13}$$

And similarly expectations are given by

$$\mathbb{E}_P[Y_X] = \int_{\Omega_Y} \int_{\Omega_X} yP(Y_x = y \mid X = x)P(X = x)dxdy. \tag{14}$$

In turn, which action is optimal is represented by a random variable $\tilde{X} : \Omega_{\boldsymbol{\xi}} \times \Omega_{\boldsymbol{\theta}} \mapsto \Omega_X$ where $\Omega_{\boldsymbol{\xi}} \times \Omega_{\boldsymbol{\theta}}$ defines the space of possible models $M := M(\boldsymbol{\xi}, \boldsymbol{\theta}) \in \mathcal{M}(\mathcal{G}_{\pi*})$ that parameterize the deployment environment $\pi^*$. For example, $P_{M \sim \mathcal{M}(\mathcal{G}_{\pi*})}(\tilde{X} = x) = P_{M \sim \mathcal{M}(\mathcal{G}_{\pi*})}(\mathbb{E}_{P_M}[Y_x] > \mathbb{E}_{P_M}[Y_{x'}], \forall x' \in \Omega_X \backslash \{x\})$ where $P_{M \sim \mathcal{M}(\mathcal{G}_{\pi*})}$ is a probability mass function defined over the space of possible models $\mathcal{M}(\mathcal{G}_{\pi*})$.

**Information Theory Definitions**  This section provides a definition of all the information-theoretic terms used throughout and shows that they can be generalized to mixed quantities that include discrete and continuous random variables in a consistent manner. For a discrete random variable $X \in \Omega_X$, the Shannon entropy and its conditional counter parts are defined as

$$\mathcal{H}(X) := - \sum_{x \in \Omega_X} P(X = x) \log P(X = x),$$

$$\mathcal{H}(X \mid Z = z) := - \sum_{x \in \Omega_X} P(X = x \mid Z = z) \log P(X = x \mid Z = z) \text{ and,}$$

$$\mathcal{H}(X \mid Z) := \sum_{z \in \Omega_Z} \mathcal{H}(X \mid Z = z)P(Z = z).$$

With a slight abuse of notation, we will define the *filtered entropy* as

$$\mathcal{H}(X \mid \bar{\boldsymbol{v}}) := - \sum_{x \in \mathcal{X}} P(X = x \mid \bar{\boldsymbol{v}}) \log P(X = x \mid \bar{\boldsymbol{v}}),$$

where $\bar{\boldsymbol{v}}$ is a dataset.

Given two probability measures $P$ and $Q$, where $P$ is absolutely continuous w.r.t. $Q$ so that the Radon-Nikodym derivative $\frac{dP}{dQ}$ is well defined, the Kullback-Leibler divergence is

$$\mathcal{D}_{KL}(P\|Q) := \int \log \frac{dP}{dQ} dP,$$

which allows the standard definition of *mutual information*:

$$I_P(X;Y) := \mathcal{D}_{KL}(P(X,Y)\|P(X)P(Y)),$$

where the subscript $P$ that denotes the probability distribution used is sometimes omitted if unambiguous. Equivalently, we can define the mutual information and *conditional mutual information* by

$$I(X;Y) = \mathcal{H}(X) - \mathcal{H}(X \mid Y) \text{ and } I(X;Y \mid Z) = \mathcal{H}(X \mid Z) - \mathcal{H}(X \mid Y, Z),$$

and we will also need the *filtered mutual information*,

$$I_{P(\cdot \mid Z=z)}(X;Y) := \mathcal{D}_{KL}(P(X,Y \mid Z = z) \| P(X \mid z)P(Y \mid Z = z)),$$

where we will typically evaluate $I_{P(\cdot \mid \bar{V}_t, \bar{v})}(X;Y) := D(P(X,Y \mid \bar{V}_t, \bar{v}) \| P(X \mid \bar{V}_t, \bar{v})P(Y \mid \bar{V}_t, \bar{v}))$ where recall $\bar{V}_t := \{V_{X^{(1)}}, \ldots, V_{X^{(t-1)}}\}$ denotes the random variables associated with the agent's history of interactions with $\pi^*$ up to round $t$. The conditional mutual information is related to the filtered mutual information by an expectation: $\mathbb{E}[I_{P(\cdot \mid Z)}(X;Y)] = I(X;Y \mid Z)$, with respect to $P(Z)$. We will frequently use this fact to show that

$$\mathbb{E}[I_{P(\cdot \mid \bar{V}_t, \bar{v})}(X;Y)] = I_{P(\cdot \mid \bar{v})}(X;Y \mid \bar{V}_t),$$

where the expectation is taken with respect to $P(\bar{V}_t \mid \bar{v})$. Perhaps the most important property of mutual information will be the chain rule,

$$I(X; Z_1, \ldots, Z_m) = I(X; Z_1) + I(X; Z_2 \mid Z_1) + \ldots + I(X; Z_m \mid Z_1, \ldots, Z_{m-1}).$$

We will need to evaluate $I(\boldsymbol{\theta}, \boldsymbol{\xi}; Z)$, where $Z$ is a discrete random variable, *e.g.* $\bar{v}_t$ or $V^{(t)}$, $\boldsymbol{\theta}$ is continuous, and $\boldsymbol{\xi}$ is discrete. We can evaluate this mutual information by writing

$$\begin{aligned}
I(\boldsymbol{\theta}, \boldsymbol{\xi}; Z) &= I(Z; \boldsymbol{\theta}, \boldsymbol{\xi}) \\
&= I(Z; \boldsymbol{\theta}) + I(Z; \boldsymbol{\xi} \mid \boldsymbol{\theta}) \\
&= \mathcal{H}(Z) - \mathcal{H}(Z \mid \boldsymbol{\theta}) + \mathcal{H}(Z \mid \boldsymbol{\theta}) - \mathcal{H}(Z \mid \boldsymbol{\xi}, \boldsymbol{\theta}),
\end{aligned}$$

which are all well defined.

## C.2  Proofs

For readability, we restate all propositions.

**Proposition 1** (restated). *Let $R_T$ denote the regret incurred by following Thompson sampling (Alg. 1). For any $T \in \mathbb{N}$ and $\Gamma \geqslant \Gamma_t$, then*

$$\mathbb{E}[R_T \mid \bar{v}] \leqslant \Gamma \sqrt{\mathcal{H}(\tilde{X} \mid \bar{v})T},$$

*where $\mathcal{H}(\tilde{X} \mid \bar{v})$ is the conditional entropy of $\tilde{X}$ given $\bar{v}$.*

*Proof.* This analysis extends [35, Prop. 1] to account for the case in which prior data is conditioned upon. We first bound the expected regret by using the law of total expectations and introducing the mutual information between the optimal action and the action observation tuple,

$$\mathbb{E}[R_T \mid \bar{v}] = \mathbb{E}\left[\sum_{t=1}^{T} Y_{\tilde{X}} - Y_{X^{(t)}} \mid \bar{v}\right] \tag{15}$$

$$= \mathbb{E} \sum_{t=1}^{T} \mathbb{E}\left[Y_{\tilde{X}} - Y_{X^{(t)}} \mid \bar{V}_t, \bar{v}\right] \tag{16}$$

$$= \mathbb{E} \sum_{t=1}^{T} \sqrt{I_{P(\cdot \mid \bar{V}_t, \bar{v})}(\tilde{X}; V_{X^{(t)}})} \frac{\mathbb{E}\left[Y_{\tilde{X}} - Y_{X^{(t)}} \mid \bar{V}_t, \bar{v}\right]}{\sqrt{I_{P(\cdot \mid \bar{V}_t, \bar{v})}(\tilde{X}; V_{X^{(t)}})}}. \tag{17}$$

By using the KL divergence definition of mutual information and using the fact that Thompson sampling precisely chooses actions according the their probability of being optimal, it can be shown that,

$$\begin{aligned}
\mathbb{E}\left[Y_{\tilde{X}} - Y_{X^{(t)}}\right] &= \sum_{x \in \Omega_X} P(\tilde{X} = x)\mathbb{E}[Y_x \mid \tilde{X} = x] - \sum_{x \in \Omega_X} P(X = x)\mathbb{E}[Y_x] \\
&= \sum_{x \in \Omega_X} P(\tilde{X} = x)(\mathbb{E}[Y_x \mid \tilde{X} = x] - \mathbb{E}[Y_x]),
\end{aligned}$$

where $P(X = x) = P(\tilde{X} = x)$ by definition of the policy used by Thompson sampling. Let the information ratio $\Gamma_t$ is defined as an upperbound on the ratio

$$\frac{\mathbb{E}\left[Y_{\tilde{X}} - Y_{X^{(t)}} \mid \bar{V}_t, \bar{v}\right]}{\sqrt{I_{P(\cdot|\bar{V}_t,\bar{v})}(\tilde{X}; V_{X^{(t)}})}} \leqslant \Gamma_t.$$

By writing $\Gamma \geqslant \Gamma_t, \forall t$, applying this bound yields

$$\mathbb{E}[R_T \mid \bar{v}] = \mathbb{E} \sum_{t=1}^{T} \sqrt{I_{P(\cdot|\bar{V}_t,\bar{v})}(\tilde{X}; V_{X^{(t)}})} \frac{\mathbb{E}\left[Y_{\tilde{X}} - Y_{X^{(t)}} \mid \bar{V}_t, \bar{v}\right]}{\sqrt{I_{P(\cdot|\bar{V}_t,\bar{v})}(\tilde{X}; V_{X^{(t)}})}}$$

$$\leqslant \mathbb{E} \sum_{t=1}^{T} \Gamma_t \sqrt{I_{P(\cdot|\bar{V}_t,\bar{v})}(\tilde{X}; V_{X^{(t)}})}$$

$$\leqslant \Gamma \mathbb{E} \sum_{t=1}^{T} \sqrt{I_{P(\cdot|\bar{V}_t,\bar{v})}(\tilde{X}; V_{X^{(t)}})}$$

$$\leqslant \Gamma \sqrt{T \mathbb{E} \sum_{t=1}^{T} I_{P(\cdot|\bar{V}_t,\bar{v})}(\tilde{X}; V_{X^{(t)}})}$$

$$\leqslant \Gamma \sqrt{T \mathcal{H}(\tilde{X} \mid \bar{v})},$$

where the second to last inequality follows from the Cauchy-Schwartz inequality and the last inequality follows from the fact that,

$$\mathbb{E} \sum_{t=1}^{T} I_{P(\cdot|\bar{V}_t,\bar{v})}(\tilde{X}; V_{X^{(t)}}) = \sum_{t=1}^{T} I_{P(\cdot|\bar{v})}(\tilde{X}; V_{X^{(t)}} \mid \bar{V}_t) \tag{18}$$

$$= I_{P(\cdot|\bar{v})}(\tilde{X}; (V_{X^{(T)}}, \ldots, V_{X^{(1)}})) \tag{19}$$

$$= \mathcal{H}(\tilde{X} \mid \bar{v}) - \mathcal{H}(\tilde{X} \mid V_{X^{(T)}}, \ldots, V_{X^{(1)}}, \bar{v}) \tag{20}$$

$$\leqslant \mathcal{H}(\tilde{X} \mid \bar{v}), \tag{21}$$

by the chain rule for the mutual information and given the non-negativity of the entropy. $\qquad \square$

**Proposition 2** (restated). *Let $R_T$ denote the regret incurred following the policy defined by Alg. 1. For any $T \in \mathbb{N}$, if Eq. (7) holds with $\Gamma \geqslant \Gamma_t$ for all $t$,*

$$\mathbb{E}[R_T \mid \bar{v}] \leqslant \Gamma \sqrt{T I_{P(\cdot|\bar{v})}(\boldsymbol{\theta}, \boldsymbol{\xi}; V_{X^{(1)}}, \ldots, V_{X^{(T)}})} + \sum_{t=1}^{T} \mathbb{E}[\epsilon_t].$$

*Proof.* This proof uses a different characterization of the per round regret to explicitly consider the influence of model parameters $(\boldsymbol{\theta}, \boldsymbol{\xi})$ on information gain. Assume instead that there exists $\Gamma_t$ such that

$$\mathbb{E}[Y_{\tilde{X}} - Y_{X^{(t)}} \mid \bar{V}_t, \bar{v}] \leqslant \Gamma_t \sqrt{I_{P(\cdot|\bar{V}_t,\bar{v})}(\boldsymbol{\theta}, \boldsymbol{\xi}; V_{X^{(t)}})} + \epsilon_t, \tag{22}$$

where $\epsilon_t > 0$ is an additional slack term. The proof strategy follows that of [25, Prop. 2] where unconditional regret bounds were shown. Given Eq. (22) the following derivation holds,

$$\mathbb{E}[R_T \mid \bar{\boldsymbol{v}}] = \mathbb{E}\left[\sum_{t=1}^{T} Y_{\bar{X}} - Y_{X^{(t)}} \mid \bar{\boldsymbol{v}}\right] \qquad \text{by definition}$$

$$= \mathbb{E}\left[\sum_{t=1}^{T} \mathbb{E}\left[Y_{\bar{X}} - Y_{X^{(t)}} \mid \bar{\boldsymbol{v}}, \bar{\boldsymbol{V}}_t\right]\right] \qquad \text{by the law of iterated expectations}$$

$$\leqslant \mathbb{E}\left[\sum_{t=1}^{T} \Gamma_t \sqrt{I_{P(\cdot \mid \bar{\boldsymbol{V}}_t, \bar{\boldsymbol{v}})}(\boldsymbol{\theta}, \boldsymbol{\xi}; \boldsymbol{V}_{X^{(t)}})} + \epsilon_t\right] \qquad \text{by definition of } \Gamma_t \text{ and } \epsilon_t$$

$$\leqslant \Gamma \sqrt{T \sum_{t=1}^{T} \mathbb{E}\left[I_{P(\cdot \mid \bar{\boldsymbol{V}}_t, \bar{\boldsymbol{v}})}(\boldsymbol{\theta}, \boldsymbol{\xi}; \boldsymbol{V}_{X^{(t)}})\right]} + \mathbb{E}\left[\sum_t \epsilon_t\right] \qquad \text{by Cauchy-Schwarz's inequality}$$

$$= \Gamma \sqrt{T \sum_{t=1}^{T} I_{P(\cdot \mid \bar{\boldsymbol{v}})}(\boldsymbol{\theta}, \boldsymbol{\xi}; \boldsymbol{V}_{X^{(t)}} \mid \bar{\boldsymbol{V}}_t)} + \mathbb{E}\left[\sum_t \epsilon_t\right] \qquad \text{by definition of the expectation of } I_P$$

$$= \Gamma \sqrt{T I_{P(\cdot \mid \bar{\boldsymbol{v}})}(\boldsymbol{\theta}, \boldsymbol{\xi}; \boldsymbol{V}_{X^{(1)}}, \ldots, \boldsymbol{V}_{X^{(T)}})} + \mathbb{E}\left[\sum_t \epsilon_t\right] \qquad \text{by the chain rule for mutual information.}$$

$\square$

**Proposition 3** (restated). *Fix $\delta > 0$ and choose $\Gamma_t$ such that $\left|Y_x - \mathbb{E}[Y_x \mid \bar{\boldsymbol{V}}_t, \bar{\boldsymbol{v}}]\right| \leqslant \frac{\Gamma_t}{2}\sqrt{I_{P(\cdot \mid \bar{\boldsymbol{V}}_t, \bar{\boldsymbol{v}})}(\boldsymbol{\theta}, \boldsymbol{\xi}; Y_x)}$ for all $x \in \Omega_X$ simultaneously with probability greater than $1 - \delta$. Then Alg. 1 chooses actions $X^{(t)}$ that satisfy*

$$\mathbb{E}[Y_{\bar{X}} - Y_{X^{(t)}} \mid \bar{\boldsymbol{V}}_t, \bar{\boldsymbol{v}}] \leqslant \Gamma_t \sqrt{I_{P(\cdot \mid \bar{\boldsymbol{V}}_t, \bar{\boldsymbol{v}})}(\boldsymbol{\theta}, \boldsymbol{\xi}; \boldsymbol{V}_{X^{(t)}})} + \delta B,$$

*where $B \geqslant 0$ is such that $\sup_{y, y' \in \Omega_Y} y - y' \leqslant B$.*

*Proof.* This proposition extends [25, Lem. 3] to account for the case in which prior data is conditioned upon. Thompson sampling, by definition, samples an action according to its probability of being optimal with the current distribution of parameter values,

$$\mathbb{E}[Y_{\bar{X}} - Y_{X^{(t)}} \mid \bar{\boldsymbol{V}}_t, \bar{\boldsymbol{v}}] = \mathbb{E}\left[\mathbb{E}_{P_{\hat{M}}}[Y_{X^{(t)}} \mid \bar{\boldsymbol{V}}_t, \bar{\boldsymbol{v}}] - \mathbb{E}_{P_{\pi*}}[Y_{X^{(t)}} \mid \bar{\boldsymbol{V}}_t, \bar{\boldsymbol{v}}]\right],$$

where $\hat{M} := M(\hat{\boldsymbol{\xi}}, \hat{\boldsymbol{\theta}})$ defines the SCM that is obtained with the current sample $\hat{\boldsymbol{\xi}}, \hat{\boldsymbol{\theta}} \sim P(\boldsymbol{\xi}, \boldsymbol{\theta} \mid \bar{\boldsymbol{v}}_t, \bar{\boldsymbol{v}})$ from the posterior distribution at round $t$, while $M_{\pi*}$ refers to the true underlying SCM of the environment. Expectations with respect to the distributions $P_{\hat{M}}$ and $P_{\pi*}$ are conditioned on a specific model of the environment. Define,

$$\mathcal{E} := \left\{(\boldsymbol{\xi}, \boldsymbol{\theta}) \in \Omega_{\boldsymbol{\xi}} \times \Omega_{\boldsymbol{\theta}} : \left|Y_{x, M(\boldsymbol{\xi}, \boldsymbol{\theta})} - \mathbb{E}_{M \sim P_{\mathcal{M}(\mathcal{G}_{\pi*})}}[Y_{x, M}]\right| \leqslant \Gamma_t \sqrt{I_{P(\cdot \mid \bar{\boldsymbol{V}}_t, \bar{\boldsymbol{v}})}(\boldsymbol{\theta}, \boldsymbol{\xi}; Y_x)}, \forall x \in \Omega_X\right\},$$

where $Y_{x, M(\boldsymbol{\xi}, \boldsymbol{\theta})}$ refers to the random variable $Y$ in SCM $M(\boldsymbol{\xi}, \boldsymbol{\theta})$ in which we intervene to set $X$ to $x$, and $Y_{x, M}$ refers to the random variable $Y$ in SCM $M$ in which we set $X$ to $x$.

The probability matching property of Thompson sampling implies that $P(\hat{\boldsymbol{\xi}}, \hat{\boldsymbol{\theta}} \in \mathcal{E} \mid \bar{\boldsymbol{v}}_t, \bar{\boldsymbol{v}}) \geqslant 1 - \delta/2$. It follows then, using the same proof strategy as [25, Lemma 3], that

$$
\begin{aligned}
\mathbb{E}[Y_{\tilde{X}} - Y_{X^{(t)}} \mid \bar{\boldsymbol{V}}_t, \bar{\boldsymbol{v}}] &\leqslant \mathbb{E}_{P(\cdot \mid \bar{\boldsymbol{V}}_t, \bar{\boldsymbol{v}})} \left[ \mathbb{1}\{(\hat{\boldsymbol{\xi}}, \hat{\boldsymbol{\theta}}), (\boldsymbol{\xi}, \boldsymbol{\theta}) \in \mathcal{E}\}(Y_{X^{(t)}, \hat{M}} - Y_{X^{(t)}, M}) \right] + \delta B \\
&\leqslant \mathbb{E}_{P(\cdot \mid \bar{\boldsymbol{V}}_t, \bar{\boldsymbol{v}})} \left[ \Gamma_t \sum_{x \in \Omega_X} \mathbb{1}\{X^{(t)} = x\} \sqrt{I_{P(\cdot \mid \bar{\boldsymbol{V}}_t, \bar{\boldsymbol{v}})}(\boldsymbol{\theta}, \boldsymbol{\xi}; Y_x)} \right] + \delta B \\
&\leqslant \Gamma_t \sum_{x \in \Omega_X} P(X^{(t)} = x \mid \bar{\boldsymbol{V}}_t, \bar{\boldsymbol{v}})) \sqrt{I_{P(\cdot \mid \bar{\boldsymbol{V}}_t, \bar{\boldsymbol{v}})}(\boldsymbol{\theta}, \boldsymbol{\xi}; Y_x)} + \delta B \\
&\leqslant \Gamma_t \sqrt{\sum_{x \in \Omega_X} P(X^{(t)} = x \mid \bar{\boldsymbol{V}}_t, \bar{\boldsymbol{v}})) I_{P(\cdot \mid \bar{\boldsymbol{V}}_t, \bar{\boldsymbol{v}})}(\boldsymbol{\theta}, \boldsymbol{\xi}; Y_x)} + \delta B \\
&= \Gamma_t \sqrt{\sum_{x \in \Omega_X} P(X^{(t)} = x \mid \bar{\boldsymbol{V}}_t, \bar{\boldsymbol{v}})) I_{P(\cdot \mid \bar{\boldsymbol{V}}_t, \bar{\boldsymbol{v}})}(\boldsymbol{\theta}, \boldsymbol{\xi}; Y_{X^{(t)}} \mid X^{(t)} = x)} + \delta B \\
&= \Gamma_t \sqrt{I_{P(\cdot \mid \bar{\boldsymbol{V}}_t, \bar{\boldsymbol{v}})}(\boldsymbol{\theta}, \boldsymbol{\xi}; Y_{X^{(t)}} \mid X^{(t)})} + \delta B \\
&= \Gamma_t \sqrt{I_{P(\cdot \mid \bar{\boldsymbol{V}}_t, \bar{\boldsymbol{v}})}(\boldsymbol{\theta}, \boldsymbol{\xi}; Y_{X^{(t)}}, X^{(t)})} + \delta B,
\end{aligned}
$$

where the last equalities follow from the conditional independence between $X^{(t)}$ and $(\boldsymbol{\theta}, \boldsymbol{\xi})$ conditioned on $\bar{\boldsymbol{V}}_t, \bar{\boldsymbol{v}}$.

This proposition might guide us to obtain a per-period regret bound that in particular determines a choice for the noise term $\epsilon_t$, which results in an additive term in Prop. 2. A common choice for $\epsilon_t$ is $1/T$ where $T$ is the number of experimentation rounds and therefore the additive term in the regret is simply a constant. This is also the choice made by [25]. In this way, the regret bound is tuned with knowledge of the horizon, which is common in online learning. □

### C.3  Information ratio computation for specific example

The theoretical statements presented in the main body of this work only implicitly relate bounds on regret with the assumed environment discrepancies assumed in selection diagrams. It is challenging in general to provide more explicit connections. Although similarity in structure between domains is explicitly encoded in selection diagrams, the association between discrepancies and posterior distributions on parameters or reward probabilities learned from prior data for an arbitrary input selection diagram are complex. For instance, it doesn't necessarily hold that few discrepancies lead to narrow reward probability posterior distributions, and conversely it doesn't necessarily hold that large discrepancies, *e.g.* many selection nodes, lead to wide reward probability posterior distributions.

In particular, consider an extreme example in which source and target domains are equal and described by $\mathcal{G} := \{X \to Y, X \dashleftarrow\dashrightarrow Y\}$. Even with infinite data and no discrepancy, it can be shown that posterior reward probabilities $P^*(y \mid do(x))$ are bounded in the interval $[P(x, y), P(x, y) + 1 - P(x)]$ which might be arbitrarily wide depending on the underlying SCM. In contrast, in other examples with certain graphs in which source and target domains differ on an arbitrarily large set of causal mechanisms (that are, however, irrelevant for the computation of posterior reward probabilities) posterior reward probabilities could be made arbitrarily narrow with sufficient data.

The information ratio $\Gamma$ can highlight the trade-off between regret and graph structure. For specific graphs, a bound on the information ratio $\Gamma$ can be analytically computed and used to convey some of the performance gains to be expected, for intuition. Recall that the information ratio is defined as a scalar $\Gamma_t$ such that,

$$
\mathbb{E}[Y_{\tilde{X}} - Y_{X^{(t)}} \mid \bar{\boldsymbol{V}}_t, \bar{\boldsymbol{v}}] = \sqrt{\Gamma_t I_{P(\cdot \mid \bar{\boldsymbol{V}}_t, \bar{\boldsymbol{v}})} \left( \tilde{X}; \boldsymbol{V}_{X^{(t)}} \right)}, \tag{23}
$$

The information ratio can be bounded: $1/2 \leqslant \Gamma_t \leqslant |\Omega_X|/2$, and describes how much information is revealed about reward distributions when an arm is pulled: on one end $\Gamma_t = 1/2$ in problems with full information arise when the outcome $Y_x$ is perfectly revealed by observing $Y_{x'}$, and on the other end $\Gamma_t = |\Omega_X|/2$ in problems where random variables $Y_x$ and $Y_{x'}$ are independent for any

$x \neq x'$. Bounds on the information ratio have also been shown for linear bandits [35]. With our parameterization of $P_{\pi*}(y_x)$ the probability of exogenous variables encoded in $\boldsymbol{\theta}$ is shared while $\boldsymbol{\xi}$ is not, thus some information about reward distributions is expected to be gained on all arms in each MAB round, and $1/2 < \Gamma_t < |\Omega_X|/2$.

In the following we will quantify the amount of sharing between reward distributions and compute a corresponding a smaller upperbound for the information ratio for a specific graph in which the computation is tractable following the proof strategy of [35, Proposition 5]. This illustrates more precisely the regret gains that can be expected from having a shared parameterization for reward distributions.

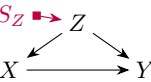

Figure 10: $\mathcal{G}^{a,*}$

We consider the selection diagram with domain-specific confounding given in Fig. 10 and the computation of $P^*(y \mid do(x)) = \sum_{z \in \Omega_Z} P^a(y \mid x, z) P^*(z)$ given knowledge of $P^a(x, y, z)$. This graph is interesting because we may directly parameterize the conditional distributions involved in $P^*(y \mid do(x))$ instead of considering its underlying SCM. Let $Y$ be binary for simplicity (all arguments hold also more generally). For the purposes of this discussion, we will assume that $P^a(y \mid x, z)$ can be approximated arbitrarily well from sufficient prior data. It holds that

$$P^*(y \mid do(x)) = \sum_{z \in \Omega_Z} P^a(y \mid x, z) P^*(z) \tag{24}$$

$$= \boldsymbol{\phi}(x)^T \boldsymbol{p}, \tag{25}$$

where $\boldsymbol{\phi}(x) \in \mathbb{R}^{|\Omega_Z|}$ is a column vector with $i$-th entry $P_{\pi^a}(Y = 1 \mid x, z = i)$ and $\boldsymbol{p} \in \mathbb{R}^{|\Omega_Z|}$ is a column vector with $i$-th entry $P(Z = i)$. The latter has some distribution in the interval $[0, 1]$ and in particular we denote its mean by $\mu := \mathbb{E}[\boldsymbol{p}]$ and its mean conditioned on the optimal action being $x_j$ by $\mu_j := \mathbb{E}[\boldsymbol{p} \mid \tilde{X} = x_j]$. Recall that $\tilde{X}$ denotes the optimal action.

Let $\alpha_i := P(\tilde{X} = x_i)$ and define $M \in \mathbb{R}^{|\Omega_X| \times |\Omega_X|}$ by its $(i, j)$-th entry,

$$M_{i,j} := \sqrt{\alpha_i \alpha_j} \left( \mathbb{E}[Y_{x_i} \mid \tilde{X} = x_j] - \mathbb{E}[Y_{x_i}] \right) \tag{26}$$

$$= \sqrt{\alpha_i \alpha_j} \left( \boldsymbol{\phi}(x_i)^T \mathbb{E}[\boldsymbol{p} \mid \tilde{X} = x_j] - \boldsymbol{\phi}(x_i)^T \mathbb{E}[\boldsymbol{p}] \right) \tag{27}$$

$$= \sqrt{\alpha_i \alpha_j} \left( \mathbb{E}[\boldsymbol{p} \mid \tilde{X} = x_j] - \mathbb{E}[\boldsymbol{p}] \right)^T \boldsymbol{\phi}(x_i). \tag{28}$$

All expectations are taken with respect to $P(\cdot \mid \bar{\boldsymbol{V}}_t, \bar{\boldsymbol{v}})$. $M$ can therefore be written as a product of two matrices of rank $|\Omega_X|$. With this definition, it was shown by [35, Proposition 5] that,

$$\mathbb{E}\left[ Y_{\tilde{X}} - Y_{X^{(t)}} \mid \bar{\boldsymbol{V}}_t, \bar{\boldsymbol{v}} \right]^2 = \text{Trace}(M), \tag{29}$$

and that,

$$I_{P(\cdot \mid \bar{\boldsymbol{V}}_t, \bar{\boldsymbol{v}})}(\tilde{X}; \boldsymbol{V}_{X^{(t)}}) \geq 2\|M\|_F^2, \tag{30}$$

leading to the fact that the information ratio,

$$\Gamma_t := \frac{\mathbb{E}\left[ Y_{\tilde{X}} - Y_{X^{(t)}} \mid \bar{\boldsymbol{V}}_t, \bar{\boldsymbol{v}} \right]^2}{I_{P(\cdot \mid \bar{\boldsymbol{V}}_t, \bar{\boldsymbol{v}})}(\tilde{X}; (X, Y_X))} \leq \frac{\text{Rank}(M)}{2} = \frac{|\Omega_Z|}{2}. \tag{31}$$

Now, $|\Omega_Z| << |\Omega_X|$ in certain applications which shows that the information ratio may be smaller than the worst-case value of $|\Omega_X|/2$ due to sharing of $P^*(z)$ across different actions $x$.

# D   Details on the data generating mechanisms

**Bow graph with domain-specific confounding.**   To generate source data we choose an SCM $M^a$ compatible with the graph specified as follows: $P(u_Z), P(u_X, P(u_Y, P(u_{XY}))$ are given by independent Gaussian distributions with mean 0 and variance 1, and each observation $(z, x, y)$ is generated from $(u_Z, u_X, u_Y, u_{XY})$ using the structural assignments: $z \leftarrow \mathbb{1}\{u_Z < 1\}, x \leftarrow \mathbb{1}\{z + u_{XY} - u_X > 0\}, y \leftarrow \mathbb{1}\{x - 0.5z + 2u_{XY} > 0\}$.

The deployment domain $\pi^*$ is given by a different SCM $M^*$ that varies in the causal mechanism relating to $Z$. It is given by the following SCM: $P(u_Z), P(u_X, P(u_Y, P(u_{XY}))$ are given by independent Gaussian distributions with mean 0 and variance 1, and each observation $(z, x, y)$ is generated from $(u_Z), u_X, u_Y, u_{XY})$ using the structural assignments: $z \leftarrow \mathbb{1}\{u_Z < 0\}, x \leftarrow \mathbb{1}\{z + u_{XY} - u_X > 0\}, y \leftarrow \mathbb{1}\{x - 0.5z + 2u_{XY} > 0\}$.

**Hypertension example.**   To generate data from the patient population $\pi^a$ we choose an SCM $M^a$ compatible with the graph specified as follows: $P(u_Z), P(u_X, P(u_W, P(u_{XY}))$ are given by independent Gaussian distributions with mean 0 and variance 1, and each observation $(z, x, w, y)$ is generated from $(u_Z), u_X, u_Y, u_{XY})$ using the structural assignments: $z \leftarrow \mathbb{1}\{u_Z > 0\}, x \leftarrow \text{int}\{0.5z + \mathbb{1}\{u_{XY} > 0\} + 2\mathbb{1}\{u_{XY} > 0\} - \mathbb{1}\{u_X > 0\} > 0\} + 2\mathbb{1}\{u_X > 1\} + 1\}, w \leftarrow \mathbb{1}\{0.3x - u_W - 0.8 > 0\}, y \leftarrow \mathbb{1}\{w - 0.5z + u_{XY} > 0\}$. $\text{int}\{\cdot\}$ stands for the integer part of the content of the brackets.

The domain $\pi^*$ in which the MAB is deployed is given by a different SCM $M^*$ that varies in the causal mechanism relating to $Z$ and $W$ with respect to $M^b$, and varies in $W$ with respect to $M^a$. It is defined by: $P(u_Z), P(u_X, P(u_W, P(u_{XY}))$ are given by independent Gaussian distributions with mean 0 and variance 1, and each observation $(z, x, w, y)$ is generated from $(u_Z), u_X, u_Y, u_{XY})$ using the structural assignments: $z \leftarrow \mathbb{1}\{u_Z > 0\}, x \leftarrow \text{int}\{0.5z + \mathbb{1}\{u_{XY} > 0\} + 2\mathbb{1}\{u_{XY} > 0\} - \mathbb{1}\{u_X > 0\} + 2\mathbb{1}\{u_X > 1\} + 1\}, w \leftarrow \mathbb{1}\{0.2x - u_W - 1.8 > 0\}, y \leftarrow \mathbb{1}\{w - 0.5z + u_{XY} > 0\}$.

**Digital advertising example.**   This example considers data from two domains. Data from the source domain $\pi^a$ is given by a SCM $M^a$ compatible with the graph that is specified as follows: $P(u_Z), P(u_X, P(u_W, P(u_{XW}, P(u_{WY}, P(u_A))$ are given by independent Gaussian distributions with mean 0 and variance 1, and each observation $(z, x, w, a, y)$ is generated from a sample $u_Z, u_X, u_W, u_{XW}, u_{WY}, u_A$ using the structural assignments: $w \leftarrow \mathbb{1}\{u_{XW} + u_{WY} > 0\}, z \leftarrow \mathbb{1}\{u_Z + w > 0\}, a \leftarrow \mathbb{1}\{u_A > 0\}, x \leftarrow \text{int}\{0.5z - 0.5a + \mathbb{1}\{u_{XW} > 0\} + 2\mathbb{1}\{u_{XW} > 0.5\} - \mathbb{1}\{u_X > 0\} + 1\}, y \leftarrow \mathbb{1}\{0.2x - 0.5a + u_{WY} - 1 > 0\}$.

The domain $\pi^*$ in which the MAB is deployed is given by a different SCM $M^*$ that varies in the causal mechanism relating to $A$ with respect to $M^a$. It is defined by: $P(u_Z), P(u_X, P(u_W, P(u_{XW}, P(u_{WY}, P(u_A))$ are given by independent Gaussian distributions with mean 0 and variance 1, and each observation $(z, x, w, a, y)$ is generated from a sample $u_Z, u_X, u_W, u_{XW}, u_{WY}, u_A$ using the structural assignments: $w \leftarrow \mathbb{1}\{u_{XW} + u_{WY} > 0\}, z \leftarrow \mathbb{1}\{u_Z + w > 0\}, a \leftarrow \mathbb{1}\{u_A > 0.5\}, x \leftarrow \text{int}\{0.5z - 0.5a + \mathbb{1}\{u_{XW} > 0\} + 2\mathbb{1}\{u_{XW} > 0.5\} - \mathbb{1}\{u_X > 0\} + 1\}, y \leftarrow \mathbb{1}\{0.2x - 0.5a + u_{WY} - 1 > 0\}$.

# E  Gibbs sampling

This section gives the derivation of all conditionals using our parameterization of causal effects for the bow graph with confounding example.

The query of interest is given by $P^*(y_x = 1)$ which is first approximated using source data $\bar{v}$ and then updated using interventional data $v_{x^{(1)}}, v_{x^{(2)}}, \dots$ collected by the MAB in the deployment environment $\pi^*$.

Its parameterization, following Corol. 1, is given by

$$P^*(y_x = 1) = \sum_{u_{xy}, u_z} \mathbb{1}\{\xi_Y^{(x,z,u_{xy})} = y\} \mathbb{1}\{\xi_Z^{(u_z)} = z\} \theta_{u_{xy}} \theta_{u_z}, \tag{32}$$

where $\xi_Y^{(x,z,u_{xy})}$ and $\theta_{u_{xy}}$ parameters are shared across source and deployment environments, while $\mathbb{1}\{\xi_Z^{(u_z)} = z\}$ and $\theta_{u_z}$ is specific to the deployment environment. We start by approximating the posterior of all relevant parameters, that is $\xi_Y^{(x,z,u_{xy})}$ and $\theta_{u_{xy}}$, given $\bar{v}$, before interacting with the deployment environment $\pi^*$. Note that prior data $\bar{v}$ is not relevant for estimating $\mathbb{1}\{\xi_Z^{(u_z)} = z\}$ and $\theta_{u_z}$ and thus we maintain uniform prior distributions over these parameters at this stage. In the following, we give the derivation of the complete conditionals over $\xi_Y^{(x,z,u_{xy})}$ and $\theta_{u_{xy}}$.

1. Sampling from $P(\bar{u}_{xy}, \bar{u}_z \mid \bar{v}, \boldsymbol{\xi}, \boldsymbol{\theta})$. Let $\bar{u}_{xy} = \{u_{xy}^{(n)}, n = 1, \dots, N\}$ and $\bar{u}_z = \{u_z^{(n)}, n = 1, \dots, N\}$ denote $N$ independent samples of $U_{xy}$ and $U_z$ respectively, one corresponding to each observation $v^{(n)} = (x^{(n)}, y^{(n)}, z^{(n)})$, where is the number of prior data samples. The complete conditional can be derived following the functional dependencies in the underlying SCM given by the causal graph,

$$P(u_{xy}^{(n)}, u_z^{(n)} \mid \bar{v}, \boldsymbol{\xi}, \boldsymbol{\theta}) = P(u_{xy}^{(n)}, u_z^{(n)} \mid v^{(n)}, \boldsymbol{\xi}, \boldsymbol{\theta}) \propto P(u_{xy}^{(n)}, u_z^{(n)}, v^{(n)} \mid \boldsymbol{\xi}, \boldsymbol{\theta})$$
$$= P(y^{(n)} \mid x^{(n)}, z^{(n)}, u_{xy}^{(n)}) P(x^{(n)} \mid z^{(n)}, u_{xy}^{(n)}) P((z^{(n)} \mid u_z^{(n)}) P(u_z^{(n)}) P(u_{xy}^{(n)})$$
$$= \mathbb{1}\{\xi_Y^{(x^{(n)}, z^{(n)}, u_{xy})} = y^{(n)}\} \mathbb{1}\{\xi_X^{(z^{(n)}, u_{xy}^{(n)})} = x^{(n)}\} \mathbb{1}\{\xi_Z^{(u_z^{(n)})} = (z^{(n)}\} \theta_{u_{xy}^{(n)}} \theta_{u_z^{(n)}},$$

where we have replaced the probabilities with the corresponding parameters that are used to define them.

2. Sampling from $P(\xi_Y^{(x,z,u_{xy})} \mid \bar{v}, \bar{u}, \boldsymbol{\theta})$. Similarly, for fixed $x, z, u_{xy}$, parameter $\xi_Y^{(x,z,u_{xy})}$ is mutually independent of any other parameter in $\boldsymbol{\xi}$ given $\bar{v}, \bar{u}, \boldsymbol{\theta}$ and can be sampled separately. Recall that by definition of the underlying SCM $\xi_Y^{(x,z,u_{xy})}$ represent a deterministic mapping between inputs $x, z, u_{xy}$ and output $y \in \Omega_Y$. The value $\xi_Y^{(x,z,u_{xy})} \in \Omega_Y$ is therefore implicitly determined by the current values $\bar{v}, \bar{u}$: if there exists a tuple $(x^{(n)} = x, z^{(n)} = z, u_{xy}^{(n)} = u_{xy}, y^{(n)} = y)$ for some $n = 1 \dots, N$, then by definition $\xi_Y^{(x,z,u_{xy})} := y$ with probability 1. If no such tuple exist, then the distribution of $\xi_Y^{(x,z,u_{xy})}$ remains uniform over its domain $\Omega_Y$ as none of the data points carries information as to the mapping $(x, z, u_{xy}) \mapsto y$.

3. Sampling from $P(\theta_{u_{xy}} \mid \bar{v}, \bar{u}, \boldsymbol{\theta})$. The conditional distribution over $\theta_{u_{xy}}$ given $\bar{v}, \bar{u}$ is given by a Dirichlet distribution following the conjugacy of it with regard to categorical distributions of $U_{xy}$ and can be updated by adding the counts of each outcome $\bar{u}_{xy}$ of $U_{xy}$ in the current sample to the corresponding prior concentration parameters of the Dirichlet prior distribution.

This process eventually forms a chain of samples from the correct posterior distribution of each parameter. At this stage, the MAB is deployed in $\pi^*$ and all parameters may be updated both with prior data $\bar{v}$ as well as with additional samples $v_x$ collected online in every round of experimentation. Offline and online data points are different in kind, and contribute to updating parameters differently. For shared parameters, both types of data may be used while for parameters specific to $\pi^*$ only newly collected interventional data samples will be relevant in posterior computations.

$\xi_Y^{(x,z,u_{xy})}$ is updated using the posterior $P(\xi_Y^{(x,z,u_{xy})} \mid \bar{v}, v_{x^{(1)}}, \dots, v_{x^{(t-1)}}, \bar{u}, u_{x^{(1)}}, \dots, u_{x^{(t-1)}})$, while $\xi_Z^{(u_z)}$ is updated using the posterior $P(\xi_Z^{(u_z)} \mid \bar{v}, v_{x^{(1)}}, \dots, v_{x^{(t-1)}}, \bar{u}, u_{x^{(1)}}, \dots, u_{x^{(t-1)}}) =$

$P(\xi_Z^{(u_z)} \mid \boldsymbol{v}_{x^{(1)}}, \dots, \boldsymbol{v}_{x^{(t-1)}}, \boldsymbol{u}_{x^{(1)}}, \dots, \boldsymbol{u}_{x^{(t-1)}})$ using the same intuition as described above. Similarly, $\theta_{u_{xy}}$ is updated using the posterior $P(\theta_{u_{xy}} \mid \bar{\boldsymbol{u}}, \boldsymbol{u}_{x^{(1)}}, \dots, \boldsymbol{u}_{x^{(t-1)}})$ while $\theta_{u_z}$ is updated using the posterior $P(\theta_{u_z} \mid \bar{\boldsymbol{u}}, \boldsymbol{u}_{x^{(1)}}, \dots, \boldsymbol{u}_{x^{(t-1)}}) = P(\theta_{u_z} \mid \boldsymbol{u}_{x^{(1)}}, \dots, \boldsymbol{u}_{x^{(t-1)}})$.

