# OpenReview forum: "Transportability for Bandits with Data from Different Environments"
_NeurIPS.cc/2023/Conference — NeurIPS 2023 poster_

### Official Review · Reviewer_eikn · 2023-06-26

**Soundness:** 3 good
**Presentation:** 3 good
**Contribution:** 3 good
**Rating:** 6
**Confidence:** 1

**Summary:**

The paper analyses how bandits can exploit causal similarities (need not be fully known functionally) across different environments to improve their regret bounds. In particular, previously collected data from related environments can improve learning.

**Strengths:**

Strengths
1. Use of domain discrepancy and selection diagrams to formally treat the different environments.
2. The proposed tTS conforms nicely with how a bandit can use priors and recently tried arms to inform its posteriors.

**Weaknesses:**

None

**Questions:**

None

**Limitations:**

None identified.

---

> ### Author Rebuttal · Authors · 2023-08-09
>
> Thank you for the positive assessment of our work. We would be happy to provide any clarification that could help further with the evaluation.

---

> > ### Comment · Reviewer_eikn · 2023-08-10
> > **Thank you**
> >
> > I have looked at the other reviews and responses. It would be great if you add comments addressing the weaknesses identified by the other reviews, in addition to the answers to their questions. Thanks again for many of the clarifications.

---

> > > ### Author Response · Authors · 2023-08-11
> > > **Further comments**
> > >
> > > Thank you for engaging with us. We can certainly comment on mentioned weaknesses; in the following, we will consider each one in order.
> > >
> > > 1. "***This paper is rather notation-heavy, and it is a bit hard for readers not familiar with the language used therein.***"
> > >
> > > The notation we introduce follows existing literature in causal inference [3, 38] for the transportability formalism and existing literature on bandits [19, 29] for the Bayesian regret guarantees. We believe that careful distinctions between data generating mechanisms and distributions over variables in different environments (including their parameterization) are necessary to accurately describe the transfer learning problem. In applications, we expect that there could be range of possible settings that must be accounted for, e.g. different graphs and multiple environments each with different sets of pairwise invariances and discrepancies. Our methods and theory aim to consistently describe all of these variations and to be generally applicable in a way that is agnostic to the specific data and assumptions provided. As a result, we do acknowledge that there is a correspondingly heavier notation than found in (causal) bandit papers that do not consider learning from multiple environments. In our view, unfortunately, the introduced notation cannot be made lighter without loss of generality.
> > >
> > > 2. "***The proposed method introduces an additional computational burden.***"
> > >
> > > Learning from additional data sources involves inference of parameters given prior data and therefore results, for equal number of online runs, in an increased computational cost compared to ``online only'' algorithms. An important motivation for the use of prior source data in the first place, however, is the expectation that it could improve the efficiency of online experimentation. In particular, we show that if source environments are sufficiently related to the target environment many fewer online runs are typically necessary to attain a given level of performance, see e.g. Experiment 1 (Sec. 5). This could effectively lower the overall computational cost of the method, especially if online runs are considered to be more expensive that offline runs which might be reasonably true in applications. In any case, the proposed Thompson sampling remains a fast algorithm that could be generally applied in small to moderately-sized environments.
> > >
> > > 3. "***This work relies on having quite detailed information from the related environments– the SCM and the selection diagrams for each environment. Having this type of information is somewhat unrealistic. Nevertheless, the contribution of the paper is conceptual/theoretical and may pave the way for future work that requires less detailed information.***"
> > >
> > > For prior data to consistently inform reward distributions in the target environment, some knowledge of commonalities and differences across environments is necessary. Selection diagrams (and their encoding of strict invariances of causal mechanisms) is one type of domain knowledge that our theory leverages and under which consistent improvements can be guaranteed. This form of domain knowledge is often available in practice, especially in the medical domain or advertising where practitioners often have an understanding of the underlying biology and user characteristics, respectively. Understandably, this form of domain knowledge may be less realistic in other applications. There are two observations that could be made, however. First, some degree of mis-specification can be allowed (under which improvements could still be guaranteed) as described in Appendix B.1. Second, some relaxations of the transportability paradigm, i.e. that specifies strict equalities or inequalities between causal mechanisms, can be naturally handled by our current framework: in particular prior knowledge intervals for probability values in the target environment, as exemplified in Appendix B.1. We believe these settings do cover the kind of domain knowledge that could available in several relevant applications and for which prior data could then be used consistently to improve inference. Relaxing the graphical assumptions, e.g. by leveraging instead sets of potential causal graphs that could be learned with a causal discovery step, could be an exciting future research direction.
> > >
> > > 4. "***Furthermore, it is also somewhat unrealistic that any of the relationships between the variables in the related environment are exactly the same as the relationship as relationships in the target environment.***"
> > >
> > > This point was answered explicitly in the response to Reviewer KpW6. Let us know if we can provide further details.
> > >
> > > 5. "***Experiments seem to be conducted on toy data only. How useful is the method in practice, on real data?***"
> > >
> > > This point was answered explicitly in the response to Reviewer QDCf. Let us know if we can provide further details.

---

> ### Author Response · Authors · 2023-08-14
> **Follow-up on exchange**
>
> Dear Reviewer eikn,
>
> We appreciate you taking the time to engage with us. We were wondering whether our follow-up on the rebuttal sufficiently addressed the points you wished to have discussed in more depth. If not, we would be happy to expand on any remaining concern.
>
> Thanks again,
>
> Authors of #13757

---

### Official Review · Reviewer_QDCf · 2023-07-03

**Soundness:** 3 good
**Presentation:** 3 good
**Contribution:** 3 good
**Rating:** 6
**Confidence:** 3

**Summary:**

A framework is presented so that bandits can use data from different environments, by exploiting the causal relationship between those environments.

**Strengths:**

The contribution and problem statement are clear.

**Weaknesses:**

Experiments seem to be conducted on toy data only. How useful is the method in practice, on real data?

**Questions:**

cf above

**Limitations:**

yes

---

> ### Author Rebuttal · Authors · 2023-08-09
>
> Thank you for your review. We address your question below. We are happy to engage further to address any remaining concerns on the practical usefulness of our method.
>
> 1. ”***Experiments seem to be conducted on toy data only. How useful is the method in practice, on real data?***”
>
> To our knowledge, it is typical in the literature to work with a synthetic set up as bandits require active experimentation. We believe that the environments discussed in the experiments and their motivation, e.g. inspired by the literature on clinical trials and advertising, are realistic, and expect therefore that a similar analysis could be conducted with real data. In particular, if the selection diagram is well-specified (under mild mispecification, Appendix B.1) we expect the use of offline data to enable the proposed approach to outperform bandit algorithms such as Thompson sampling (Sec. 3) in every application.

---

> ### Author Response · Authors · 2023-08-14
> **Follow-up on rebuttal**
>
> Dear Reviewer QDCf,
>
> We are reaching the end of the discussion period. We were hoping to understand whether our rebuttal clarified your concerns or whether we could give any additional details. We would be happy to expand on our response if needed.
>
> We appreciate your time and attention. Thanks!
>
> Authors of #13757

---

### Official Review · Reviewer_KpW6 · 2023-07-04

**Soundness:** 3 good
**Presentation:** 3 good
**Contribution:** 3 good
**Rating:** 6
**Confidence:** 3

**Summary:**

This paper considers the problem of leveraging data from many different environments to warm start a bandit algorithm. The paper assumes that the environments share the same variables, and the relationships between variables in related environments can be captured by structural causal models (SCMs). Taking a Bayesian perspective, the paper defines a probability distribution over unknown quantities in the target SCM and constrains the probability distribution using the information obtained from the related environments. This probability distribution is then used to warm-start a Thompson sampling algorithm.

**Strengths:**

1. This paper has interesting theoretical contributions. The authors provide an algorithm that leverages data from related environments and are able to prove a sub-linear regret bound and the regret bound depends on a term that captures how informative the related environments are for the target environment.

2. Compelling experimental results.

3. The paper is overall clear and well-written.


**Weaknesses:**

1. This work relies on having quite detailed information from the related environments– the SCM and the selection diagrams for each environment. Having this type of information is somewhat unrealistic. Nevertheless, the contribution of the paper is conceptual/theoretical and may pave the way for future work that requires less detailed information.

2. Furthermore, it is also somewhat unrealistic that any of the relationships between the variables in the related environment are exactly the same as the relationship as relationships in the target environment. Would it be possible to place looser restrictions from the prior data than the equality constraints in Eq 2?


**Questions:**

1. In the motivating example in lines 42-60, it would be helpful if the authors could add a comment that this problem arises due to the fact that the clinical trial population and historical data population differ across observables (or unobservables).

2. [Nit] $S_{Z}$ is not defined in the text until Section 2 but appears in Figure 1, which is referenced in Section 1.

3. Would it be possible for the authors to contextualize their perspective on distribution shift within the broader literature on generalizability/transportability (e.g., Stuart, et. al. 2011, Tipton, et. al, 2013, Tipton et. al., 2014)? For example, can the different environments vary across unobservable attributes (confounders), or can they only differ in observable attributes?


References:

Stuart, Elizabeth A., et al. "The use of propensity scores to assess the generalizability of results from randomized trials." Journal of the Royal Statistical Society: Series A (Statistics in Society) 174.2 (2011): 369-386.

Tipton, Elizabeth. "Improving generalizations from experiments using propensity score subclassification: Assumptions, properties, and contexts." Journal of Educational and Behavioral Statistics 38.3 (2013): 239-266.

Tipton, Elizabeth. "How generalizable is your experiment? An index for comparing experimental samples and populations." Journal of Educational and Behavioral Statistics 39.6 (2014): 478-501.


**Limitations:**

Yes

---

> ### Author Rebuttal · Authors · 2023-08-09
>
> Thank you for your thoughtful review and constructive comments. In the following, we address each point separately and hope to clarify all concerns that were raised. Please let us know if any issues remain.
>
> 1. ”***It is also somewhat unrealistic that any of the relationships between the variables in the related environment are exactly the same as the relationships in the target environment. Would it be possible to place looser restrictions from the prior data than the equality constraints in Eq 2?***”
>
> Relaxations to the formalism in Eq. (2) are possible. One relevant example are settings in which instead it is plausible to assume that some of the causal mechanisms or probabilities are known to lie in some non-trivial interval. As each probability relates directly to some combination of model parameters, this constraint could be incorporated in posterior approximations. We discuss such a scenario in more details on lines 558-570 of the Appendix. There, specifically, instead of an equality across causal mechanisms that implies that $P^*(z) = P^a(z)$ we pose a looser restriction, e.g., $P^*(z) \in I = [P^a(z) - 0.1, P^a(z) + 0.1]$. That is, $P^*(z) = \sum_{u_z}\mathbf 1(\xi_Z^{(u_z)}=z)\theta_{u_z} \in I$, where $\mathbf 1(\cdot)$ denotes the indicator function, which defines a constraint on possible parameter values and is implemented with a rejection step while sampling from the posterior.
>
> 2. ”***In the motivating example in lines 42-60, it would be helpful if the authors could add a comment that this problem arises due to the fact that the clinical trial population and historical data population differ across observables (or unobservables).***”
>
> This observation is stated in the sentence starting line 50 and will be emphasized.
>
> 3. ”***Would it be possible for the authors to contextualize their perspective on distribution shift within the broader literature on generalizability/transportability (e.g., Stuart, et. al. 2011, Tipton, et. al, 2013, Tipton et. al., 2014)? For example, can the different environments vary across unobservable attributes (confounders), or can they only differ in observable attributes?***”
>
> Thank you for sharing these references, which we read with interest. Based on our reading, under two assumptions on the dependence of potential outcomes on treatment and domain indicators, propensity scores can be used to correct for distribution shift between target and source populations. Within the perspective of transportability (that uses selection diagrams to encode assumptions on differences and similarities between populations) this could be seen as a special case for which selection diagrams imply the independence assumptions. In general, selection diagrams may not imply this set of independencies and different weights may be applicable to correctly adjust for distribution shift. Further, we consider a generalization of this setting, so called partial transportability. In this generalization, the correct adjustment for distribution shift might not be uniquely identifiable and the goal, instead, is to infer a non-trivial interval for outcome distributions under intervention that could nevertheless provide some information to improve inference (Sec. 2.1).
>
> Selection diagrams may be used to encode differences in observable and unobservable attributes.

---

> ### Author Response · Authors · 2023-08-14
> **Follow-up on rebuttal**
>
> Dear Reviewer KpW6,
>
> With the discussion period coming to its end, we were wondering whether you had a chance to check our rebuttal. We hope to have answered all concerns to your satisfaction. If not, please don't hesitate to get in touch if there is any concern we could still help to clarify.
>
> Thank you again for your time and attention.
>
> Authors of #13757

---

> > ### Comment · Reviewer_KpW6 · 2023-08-16
> >
> > Thank you for your response! I am still reviewing this paper and the rebuttal and will provide a full response in the next day.

---

> > > ### Comment · Reviewer_KpW6 · 2023-08-19
> > >
> > > I appreciate the effort that the authors have put into the the rebuttal! Thank you! I found the following part of the authors' rebuttal helpful:
> > >
> > > ```First, some degree of mis-specification can be allowed (under which improvements could still be guaranteed) as described in Appendix B.1. Second, some relaxations of the transportability paradigm, i.e. that specifies strict equalities or inequalities between causal mechanisms, can be naturally handled by our current framework: in particular prior knowledge intervals for probability values in the target environment, as exemplified in Appendix B.1. We believe these settings do cover the kind of domain knowledge that could available in several relevant applications and for which prior data could then be used consistently to improve inference.```
> > >
> > > I recommend acceptance and am willing to raise my score. I hope the authors can add the following comments to the final version of their paper:
> > > 1) emphasizing that their approach can handle/can be adapted to handle some degree of misspecification of the transportability paradigm
> > > 2) particular applications where relationships between target/source distributions are encoded using SCMs and selection diagrams. The authors mention biological/medical applications and advertising in their rebuttal, but it would be great to see some references.

---

> > > > ### Author Response · Authors · 2023-08-20
> > > > **Response to follow-up**
> > > >
> > > > We appreciate the suggestions and positive assessment of our work; thank you for engaging with us. The two points mentioned above will be appropriately emphasised and included in the revised document, as discussed in the rebuttal.
> > > >
> > > > In addition, we include below references of causal approaches to policy optimisation (and bandits in particular) in the contexts of clinical trials, healthcare, and advertising, that could complement our existing discussions. We note, in particular, the examples in Figs. 2, 3, 4, in [1] that encode the design of case-cohort studies and clinical trials used in the MORGAM study [2] using causal graphs for the estimation of causal effects. Separately, [3] consider the International Stroke Trial and the known causal associations between relevant features for policy optimisation. In the context of advertising, [4] describe the use of causal methods with several detailed examples. Specifically, Figs. 3, 4, and 6 in [4] show examples of causal graphs that may be defined for particular computational advertising applications. See also [5] for a bandit algorithm leveraging the computational advertising graphs in [4], and [Sec. 5.2, 6] for similar causal treatments also in the context of advertising.
> > > >
> > > > [1] Karvanen, Juha. "Study design in causal models." Scandinavian Journal of Statistics 42.2 (2015): 361-377.
> > > >
> > > > [2] Evans, Alun, et al. "MORGAM (an international pooling of cardiovascular cohorts)." International journal of epidemiology 34.1 (2005): 21-27.
> > > >
> > > > [3] Kallus, Nathan, and Angela Zhou. "Confounding-robust policy improvement." Advances in neural information processing systems 31 (2018).
> > > >
> > > > [4] Bottou, Léon, et al. "Counterfactual Reasoning and Learning Systems: The Example of Computational Advertising." Journal of Machine Learning Research 14.11 (2013).
> > > >
> > > > [5] Lu, Yangyi, et al. "Regret analysis of bandit problems with causal background knowledge." Conference on Uncertainty in Artificial Intelligence. PMLR, 2020.
> > > >
> > > > [6] Sen, Rajat, et al. "Identifying best interventions through online importance sampling." International Conference on Machine Learning. PMLR, 2017.

---

### Official Review · Reviewer_DogX · 2023-07-07

**Soundness:** 3 good
**Presentation:** 2 fair
**Contribution:** 3 good
**Rating:** 6
**Confidence:** 3

**Summary:**

This paper considers the online bandit problem with
additional batch/observational data, where
the additional data could be from different (but related)
environments. The authors
present a representation of the interventional distribution, based on which one can sample from the posterior distribution
of the SCMs and select an action based on the realized model.
It is shown that the resulting regret is sublinear, and
the improvement upon an algorithm without using prior data is explicitly
dependent on the "relevance" of the other environments.

**Strengths:**

1. This paper considers an interesting problem: how to leverage observational
data to improve the online bandit algorithm.
2. The proposed method makes use of prior data in a clean way, and the
theoretical result shows an explicit dependence on the relevance of
the environments from which the prior datasets are generated, while
the algorithm itself is agnostic to this knowledge.


**Weaknesses:**

1. This paper is rather notation-heavy, and it is a bit hard
for readers not familiar with the language used therein.
2. The proposed method introduces an additional computational burden.

**Questions:**

1. I was wondering how the computation time scales with the cardinality of different variables. Appendix B.2 has briefly mentioned this --- I am curious if it would be possible to have an analytical result.
2. It might also be helpful to compare the computational time of different methods
in the simulations (in addition to the ones presented in Appendix B.2).

Minor:
Is there a typo in Algorithm 1? "$\mathcal{G} ** a$"

**Limitations:**

The authors have adequately addressed the limitations.

---

> ### Author Rebuttal · Authors · 2023-08-09
>
> Thank you for your review and feedback. We hope to have clarified your concerns in the following response, please let us know if you would like us to expand our discussion on any of it. Thanks for pointing out a typo in Algorithm 1!
>
> 1. ”***I was wondering how the computation time scales with the cardinality of different variables. Appendix B.2 has briefly mentioned this --- I am curious if it would be possible to have an analytical result.***”
>
> In a given iteration of the Gibbs sampler, posterior updates are done for each parameter separately so that computational time is proportional to the parameter count, approximately, which in turn is determined by the cardinality of variables as well as the structure of the graph. For a fixed graph, assuming that each update requires a small constant amount of time to compute, we could therefore establish analytically how computational time scales with the cardinality of variables. For arbitrary graphs an analytical result is in general more involved as the parameter count increases differently depending on the local structure of each variable.
>
> As an example for illustration, consider the graph $\mathcal G = (X\rightarrow Z \rightarrow Y, X\leftrightarrow Y)$ where $X$ is an action variable, $Z$ is a contextual variable, and $Y$ is a reward variable. Following the parameterization in Cor. 1, the cardinality of parameters is defined as follows: $|\boldsymbol{\theta}_u|= |\Omega_X|\cdot|\Omega_Z|\cdot|\Omega_Y|$, $|\boldsymbol{\xi}_X| = |\Omega_X|\cdot|\Omega_Z|\cdot|\Omega_Y|$, $|\boldsymbol{\xi}_Z| = |\Omega_X|$, $|\boldsymbol{\xi}_Y| = |\Omega_X|\cdot|\Omega_Z|\cdot|\Omega_Y|\cdot|\Omega_Z|$. We would expect run time to increase linearly with the cardinality of variables $X,Y$ and to increase “slower than quadratically” with the cardinality of $Z$.
>
> We will update the manuscript with a discussion.
>
> 2. "***It might also be helpful to compare the computational time of different methods in the simulations (in addition to the ones presented in Appendix B.2).***"
>
> We appreciate the suggestion. Over a single run of $10,000$ experimentation rounds, empirically, the run times for experiment 2 are: TS$=1.2$ seconds, UCB$= 1.7$ seconds, Random$= 0.1$ seconds, tTS$= 5.4$ seconds; and for experiment 3:  TS$=1.3$ seconds, UCB$= 1.8$ seconds, Random$= 0.1$ seconds, tTS$= 5.8$ seconds. Run times of tTS include the prior step of 1,000 rounds of sampling from the posterior distribution of parameters given 1,000 samples of prior data. We will add these analyses in the updated document.

---

> ### Author Response · Authors · 2023-08-14
> **Follow-up on rebuttal**
>
> Dear Reviewer DogX,
>
> The discussion period is reaching its end. We hope you have had the chance to check our rebuttal and wonder whether it has answered your questions. If not, we would be happy to expand on any remaining concerns.
>
> We appreciate your time and attention. Thank you!
>
> Authors of #13757

---

> > ### Comment · Reviewer_DogX · 2023-08-20
> > **response to the authors**
> >
> > I would like to thank the authors for the clarification and additional results. My concerns are addressed and I would like to maintain my score.

---

### Decision · Program_Chairs · 2023-09-21

**Decision:**

Accept (poster)

**Comment:**

The reviewers with the highest level of expertise were supportive of this paper's contributions. The following suggestions were made to improve contextualization and discussion of the results in the camera-ready version, which I strongly recommend the authors do:

- Emphasize that your approach can handle/can be adapted to handle some degree of misspecification of the transportability paradigm.
- There are particular applications where relationships between target/source distributions are encoded using SCMs and selection diagrams. The authors mention biological/medical applications and advertising in their rebuttal. It would be great to see some references for the same.